

# Annual changes of ship emissions around China under gradually promoted control policies from 2016 to 2019

Xiaotong Wang[a], Wen Yi[a], Zhaofeng Lv, Fanyuan Deng, Songxin Zheng, Hailian Xu, Junchao Zhao, Huan Liu[*], Kebin He

State Key Joint Laboratory of ESPC, School of Environment, Tsinghua University, Beijing 100084, China

*Correspondence to*: liu_env@tsinghua.edu.cn (H. Liu)

[a]*These authors contributed equally to this work*

**Abstract.** Ship emissions and coastal air pollutions around China are expected to be alleviated with the gradually implemented of domestic ship emission control (DECA) policy. However, there is so far a lack of a comprehensive post assessment on the

ship emission response after the policy implementation. This study developed a series of high spatiotemporal ship emission inventories of China's inland rivers and the 200 Nm zone from 2016 to 2019 based on an updated Ship Emission Inventory Model (SEIM v2.0) and analysed the interannual changes of emissions under the influence of both ship activity increase and gradually promoted policy. The route restoration technology in SEIM v2.0 has greatly improved the spatial distribution of ship emissions and the river vessels (RVs) are better distinguished by using the spatial frequency distribution method. From 2016

to 2019, $SO_2$ and PM emissions from ships decreased by 29.6% and 26.4%, respectively, while ship $NO_X$ emissions increased by 13.0%. Although the DECA 1.0 policy has been implemented since 2017, it was not until 2019 with the DECA 2.0 that significant emission reduction was achieved, e.g., 33.3% regarding $SO_2$. Considering the potential emissions brought by continuous growth of maritime trade, however, an even larger emission reduction effect of 39.8% was achieved in 2019 compared with the scenario without switching cleaner fuel. Although ocean-going vessels (OGVs) contributed to

approximately 2/3 of ship emissions in Chinese waters, 2/3 of them came from ships registered in other countries. Containers and bulk carriers are still the dominate contributors to ship emissions, and newly-built, large ships and ships using clean fuel oil are taking an increasingly large proportion in emission structure. The four-year consecutive daily ship emissions were presented for major ports, which timely reflects the response of step-by-step DECA policy on emissions and may provide useful references for port observation experiments and local policy making. In addition, the spatial distribution shows that a

number of ships detoured outside the scope of DECA 2.0 in 2019 to save the cost on more expensive low sulphur oil, increasing emissions in farther maritime areas. The multi-year ship emission inventory provide high-quality datasets for air quality and dispersion modellings, as well as verifications for in-situ observation experiments, which may also guide further ship emission control direction in China.



## 1 Introduction

Shipping is an important anthropogenic source of air pollutants and greenhouse gases, which has come into the view of scientists and public since the end of last century (Corbett and Fischbeck, 1997; Capaldo et al., 1999; Lawrence and Crutzen, 1999). Air pollutants emitted from ships can be further transported to inland areas by the onshore flow, along with the atmospheric chemical transformation, aggravating air pollution and endangering human health (Endresen et al., 2003; Eyring et al., 2007; Eyring et al., 2010; Corbett et al., 2007). In the past decades, despite the improvement of global fuel quality and

engine post-treatment technology, shipping emissions are still increasing driven by ever-growing maritime trade (IMO, 2020; UNCTAD, 2019). Recent study shows that global shipping emission constitutes 3% of anthropogenic $CO_2$ emissions in 2017 (IMO, 2020), and much more proportions for reactive gases, e.g., 20% to $NO_X$ and 12% to $SO_2$ emission (McDuffie et al., 2020). China, as the largest maritime trading country, sitting on seven of the world's top ten ports with even more densely distributed coastal ports, is meeting a more tough challenge due to the lagging emission control measurements compared to

European and American countries (Mao and Rutherford, 2018b).

In recent years, numerous researchers attempted to quantify ship emissions of China and evaluate their air quality impacts. These studies suggest that ship emissions of $SO_2$ in China are nearly 5 times of that from road transportation (Chen et al., 2017), and emissions within 12 nautical miles (Nm) accounts for ~ 40% of the total emissions from all ship emissions in coastal

areas (Zhang et al., 2017; Li et al., 2018). Exhaust emissions from ships contributed significantly to air pollution in coastal provinces, especially in the Yangtze River Delta (YRD), with the highest increase of annual $PM_{2.5}$ concentrations reaching up to $4 \sim 5 \mu g/m^3$ (Chen et al., 2018; Liu et al., 2018; Lv et al., 2018; Feng et al., 2019; Wang et al., 2019). During ship-plume-influenced periods, ships can even contribute 20% ~ 30 % of the total $PM_{2.5}$ concentrations (Fu and Chen, 2017; Chen et al., 2018). The adverse impact brought by ship emissions also lay huge burden on human health, causing 14,500 ~ 37,500

premature deaths in a large scale of East Asia and also hundreds of that in regional areas of Pearl River Delta (PRD) of China (Liu et al., 2016; Chen et al., 2019a).

Theses previous evaluations have made great efforts to support the formulation of China's domestic emission control area (DECA) initially designed for Bohai Rim Area (BRA), YRD and PRD and later upgraded to the whole water areas of 12 Nm

from the baseline of Mainland China, where ships entering the DECA were required to switch clean fuel oil with lower sulphur content. However, theses assessments are mostly so-called "prior assessments", namely, evaluations of the cost and benefits of environmental and health improvement by assuming control scenarios based on the earlier ship activities before the implementation of the policy. Under the circumstance of increasing shipping demand, synchronized by the step-by-step implement of control measures, the "post evaluation" is of equally importance to seek the response of actual ship activities

and emissions to the policies as well as to provide powerful foundations for in-situ observation experiments (Wu et al., 2021). Although a number of studies have demonstrated the air quality benefit due to switching low sulphur oil in local port areas



(Zhang et al., 2018a; Zhang et al., 2019b; Zou et al., 2020; Zhang et al., 2018b), there is so far a lack of a comprehensive national-scale evaluation to reflect the benefits of gradually promoted DECA policy, which is vital to guide further ship emission control direction in China.


With the advent of the big data era, characterization of ship emissions has evolved from the earliest "top-down" estimation based on global fuel consumption (Corbett et al., 1999; Endresen et al., 2003) to the "bottom-up" model based on the big data of ship automatic identification system (AIS) (Jalkanen et al., 2009; Winther et al., 2014; Liu et al., 2016; Johansson et al., 2017; Nunes et al., 2017). The AIS-based ship emission inventories have great advantages in improving the spatial-temporal

resolution for numerical simulation as well as providing possibilities of near-real-time emission estimation to meet regulatory needs (Miola and Ciuffo, 2011; Nunes et al., 2017; Huang et al., 2020). However, emission calculation methods based on big data greatly depend on the data quality, thus demanding tedious steps for data cleaning. As AIS signal loss occurs in many cases, dealing with long-time missing AIS signals has been one of the key technical problems both for scientific research and supervision (Zhang et al., 2019c; Peng et al., 2020; Zhang et al., 2020). Without targeted measures, the estimated ship emissions

would be spatially and temporally misallocated, thus further raise uncertainties of environmental impact assessment.

In this study, we developed the ship emission inventory for inland rivers and the 200 Nm zone of China from 2016 to 2019 based on the continuous global AIS data and the updated version of Shipping Emission Inventory Model (SEIM v2.0). The global AIS database including annually ~30 billion AIS signals and Ship Technical Specifications Database (STSD) covering

over 3.5 million individual vessel profiles were combined as fundamental data for emission calculation. The previous SEIM model was upgraded to SEIM v2.0 through the following three improvements: 1) developing a route restoration module to restore the most likely trajectory for missing AIS signals; 2) identifying river vessel from AIS data based on spatial frequency distribution of ship trajectories; 3) incorporating step-by-step Chinese emission control policy with daily scale. The four-year consecutive daily ship emissions and structure were analyzed from the national to port level to track the variation at a fine time

scale. The interannual spatial change of emissions from ocean-going vessels (OGVs), coastal vessels (CVs) and river vessels (RVs) were presented and compared. In addition, another scenario without the DECA policy was performed to evaluate the effect of China's gradually implemented DECA policy, considering the actual change of interannual ship activities. Results of this study provide high-quality emission inventory data for the further numerical simulation of air quality and health benefit of ship emission reduction.

## 2 Method

### 2.1 Ship emission inventory model (SEIM v2.0)

The SEIM model has been established in our previous work to develop multi-scale ship emission inventory with high spatial and temporal resolution based on a combination of satellite-based and terrestrial-based AIS data (Liu et al., 2016; Fu et al.,



2017; Liu et al., 2018). In this model, emissions were calculated based on the instantaneous operating status and power changes

for each individual ship between two successive AIS signals, usually ranging from a few seconds to a few minutes. Each active ship in AIS data was dynamically matched with its technical profiles for classification and emission calculation. With high-frequency AIS signal transmit time and geographic locations, the total emissions could be ultimately aggregated by that from all ships of all time intervals in the whole year, resulting an inventory with high temporal and spatial resolution. Technical details including the data collection and cleaning, calculation formula, emission factor adoption as well as default parameter

setting of the SEIM model has been introduced in our previous studies (Liu et al., 2016; Fu et al., 2017). Currently, SEIM considers ship emission for both air pollutants (e.g., $SO_2$, PM, $NO_X$, CO and HC) and greenhouse gases (e.g., $CO_2$, $CH_4$ and $N_2O$), from main engines, auxiliary engines and boilers.

To reduce the uncertainties of emission calculation, we introduced several techniques in the previous version of SEIM: 1) a

double-nested research domain was applied to reduce the boundary effects for regional inventories; 2) the Gradient Boosting Regression Tree (GBRT) method was adopted to estimate the default values of missing ship properties; 3) the propeller law was used to calculate the instantaneous engine loads; 4) the 10-minute linear interval interpolation method was used to figure out long-distance AIS signal gaps (Liu et al., 2016). These all contributed to improve the reliability of ship emission inventories. Here, we introduce a refined version of the SEIM v2.0 to describe the improvement of the model that applied for estimating

annual ship emission inventory around China. The major improvements include: 1) developing a route restoration module to restore the most likely trajectory for missing AIS signals; 2) distinguishing river vessel from AIS data based on spatial frequency distribution of ship trajectories; 3) incorporating step-by-step Chinese emission control policy with daily scale to timely reflect the actual emission level.

Figure 1 shows the current structure and flow chart of the SEIM v2.0, which is composed of several key modules: data pre-processing, route restoration, emission calculation, policy-abutted modification and post-processing. First, the originally collected raw AIS data and ship profile data from multiple sources are combined to form a ship activity database and STSD, and the RVs will be identified based on ship trajectories. Second, a route restoration module is applied for long-time gaps in AIS data, in which the 10-minute linear interpolation will be applied on the shorted paths instead. Third, the instantaneous

emission along with movement of ship's trajectory will be calculate based on a series of extra prepared parameters and factors. Then, the policy-abutted modification will be applied for vessel entering the DECAs to switch low sulphur fuels. Finally, the ship emission inventory datasets will be established and used for visualization and multidimensional analysis. As most of the technical methods have been described in our previous work, such as GBRT methods, emission calculation algorithm, extra parameter preparations, we focus on the study area definition, the latest data evaluations and the improvements of the SEIM

v2.0 to introduce the technical details for developing the ship emission inventory around China.



## 2.2 Study area

Ships have strong spatial mobility, unlike the on-road mobile sources that mostly have fixed geographical range of activities. Due to the complexity brought by the inconsistency of the ships' flag state, operating country and activity location, there is hardly unified standard to determine the attribution country of ship emissions. In this study, the target area for developing ship

emission inventory is the navigable inland rivers and the coastal waters approximately within 200 Nm away from the Chinese mainland's territorial sea baseline (hereinafter referred to as 200 Nm zone), as shown in Fig. 2. We defined the target area due to the following reasons. First, ship emissions occurred in this region are proved to have significant contribution to air pollution and human health of China (Lv et al., 2018), thus it is reasonable to regard China as a receptor and investigate the regional air quality impact from the surrounding ships. Second, as the current DECA is limited to 12 Nm to the baseline of territorial sea,

far less than proposed area of the international ECA (200 Nm), thus it is possible to provide a scientific reference for further scheme design by investigating the emission variation in the 200 Nm zone. In addition, the 200 Nm zone is the water region with the most intensive ship traffic and complex route, which is an appropriate demonstration area to test the effect of route restoration. The study area is also generally consistent to the research scope of other AIS-based ship emission inventory of China so as to make comparison of the corresponding results.


A double-nested domain is set to calculate ship emissions and reduce the boundary effect, in which the outer domain (D1) is 0°-90°N and 90°E-140°E and the inner domain (D2) is 14°-43°N and 104°E-130°E. The spatial distribution of emissions will be retained and presented with D2 as the boundary, and the statistical results for China will be finally made for the inland river and the 200 Nm zone. Figure 2 also shows the scope of the DECA 1.0, which include three areas, namely, BRA, YRD and

PRD (often called DECA 1.0), and the coastal areas of DECA 2.0, which is approximately equal to the area from the coastline to 12 Nm from the Chinese mainland's territorial sea baseline (hereinafter referred to as baseline). Meanwhile, ship emission within different coastal areas, i.e., from coastline to 12 Nm, 12-50 Nm, 50-100 Nm and 50-200 Nm from the baseline are also decomposed to investigate the spatial variation, which are also illustrated in Fig. 2.

## 2.3 Data pre-processing and evaluations

The global dynamic AIS data for the whole year of 2016-2019 (from January 1st to December 31st) with averagely 30 billion signals per year are collected to build a ship activity database, which provide high-frequency information including signal time, coordinate location, navigational speed and operating status, etc. The STSD has also been updated to 2019, which describes ship properties such as vessel type, dead weight tonnage (DWT) and engine power, designed speed, flag state, etc. Besides the ship data collected from Lloyd's Register and the Classification Societies of various countries, we have expanded the database

to incorporate fishing ships from Global Fishing Watch (GFW) (Kroodsma et al., 2018). As the AIS data are composed of the satellite AIS signals and the terrestrial-based AIS signals, same messages received from multiple base stations may lead to large quantities of duplicates, especially when ships are berthing. To deal with the redundant information and compress the





data size, the time spans of continuous AIS signals with their instantaneous speeds both equal to 0 and displacements less than 0.01 degree were enlarged to 10 minutes. In this way, on the premise of keeping the total operation time unchanged, the volume

of the raw AIS data was reduced. Table 1 shows the statistical results of the AIS messages and active ships for different years in this study. The increasing trend of total vessel DWT and decreasing trend of the number of identified ships operating around China indicate the improvement of average loading capacity per ship. Detailed processing method of data collection, cleaning, matching and verification are described in our previous works (Liu et al., 2016).

Figure 3 presents the statistics for dynamic activities and static technical specifications for different ships in the target region of China. As shown in Fig. 3a, the average daily operating time of all vessels in Chinese waters are approximately $5.3 \times 10^5$ hours/day. Among all vessel types, the bulk carriers operate especially long time, followed by fishing ships and containers. Most vessels show constant daily operating hours except a slightly decrease in the Spring Festival. However, fishing ships drops significantly in Summer due to the fishing-off season. Figure 3b shows the cargo fleet structure from the perspectives of

vessel number, total DWT and total installed power of main engines. In terms of the vessel numbers, the fishing ship accounts for the largest proportion of 42.5%, while general cargo also accounts for 29.8%, respectively. As for total DWT, the proportion of bulk carrier reaches 49.5%, and the oil tanker also occupies a considerable proportion (23.4%). For the total power of main engines, the proportion of container (35.4%) exceeds that of the bulk carrier (28.0%), indicating a higher engine power demand per unit volume for containers. Owning to the distinct technical specifications of different ship types, the number of vessels of

each type would not be linear with their DWT, power, navigation time, and thus emissions.

## 2.4 Model improvements

### 2.4.1 Route restoration

Even if the AIS data has high frequency to report ship activities, there are sometimes long periods of signals loss due to equipment failure or manual shutdown. This kind of signals only accounts for a minority of AIS data, but may lead to large

deviation of the amount and distribution of ship emission especially in case of long operating hours. To solve this problem, a route restoration module was developed in SEIM v2.0 to predict the most likely navigation trajectories of the lost signals and spatially reallocate ship emissions. This method has been previously experimented by Johansson et al., (2017) in a global scale. Here, we referred to their method and applied to China with more refined resolution.

The ship route restoration method is based on the Dijkstra algorithm (Cherkassky et al., 1996), which interpolates the lost signals evenly on the shortest shipping route connecting two endpoints, namely, the experiential routes. Thus, a comprehensive ship route network need to be established before applying the route restoration algorithm. As the global AIS data provide massive signals of ship locations, the historical navigation trajectories for all in-service vessels are clearly visible on map. Based on the aggregated ship traffic distribution and the geographic domain of D1 in this study, the shipping route map was





drawn and split into 870 arcs connected by 656 nodes, as depicted in Supplement Fig. S1. Regarding the shipping route map
       as an undirected graph, by applying the Dijkstra shortest-path algorithm, the shortest route path between each node-pair can
       be calculated, as well as the geodesic distance aggregated by all arcs. In this way, the ship route network connected with nodes
       and arcs were established ahead and the shortest geodesic paths for all node-pairs were pre-stored as database to look up, so
       as to improve the operation efficiency.


       Figure 4 illustrates the diagrammatic sketch of the ship route restoration algorithm, taking a segment of AIS positions as an
       example. The method can be summarized as following steps: 1) For each two consecutive AIS points $A$ and $B$, judge the
       geographical relationship between line $AB$ and the continent; 2) If line $AB$ intersects with the continent and they are not
       contained in the continent, apply the route restoration algorithm by firstly finding the nearest start node $A'$ and end node $B'$ by

traversing the pre-stored node library; 3) Look up the shortest path connecting nodes $A'$ and $B'$ (eg., $A'O_1O_2O_iO_j\cdots B'$) from
       pre-stored ship route network database and calculate average speed resulted from the geodesic distance of $D_{A'O_1O_2O_iO_j\cdots B'}$ and
       time internal $T_{AB}$; 4) For each segment $O_iO_j$ in route $A'B'$, interpolate points $p_1$, $p_2$, $p_m$, $p_n\cdots$ with time span of 600 seconds
       along the $O_iO_j$ if $T_{O_iO_j} > 600s$; 5) For each arc $p_mp_n$, calculate ship emissions based on average speed, instantaneous power
       and emission factors; 6) Calculate emissions $\sum E$ summed from each time span along the restored route.

**2.4.2 Classification of OGV, CV and RV**

       In the SEIM v2.0, vessels are classified into OGVs, CVs, and RVs for emission estimation. In China, inland vessels are having
       an increasing number with AIS equipment installed these years. As the fuel standard for RVs are more stringent compared to
       OGVs, it is necessary to distinguish them from the AIS data to calculate emissions accurately. In methodology, since OGVs
       are mostly engaged in international trade following the management of International Maritime Organization (IMO), they are

identified by meeting the condition that both valid IMO number and the Maritime Mobile Service Identify (MMSI) number
       are available. CVs and RVs are both domestic vessels designed to operate in rivers and coastal areas, respectively. However,
       in some cases, they do cross each other's navigational waters when the inland waterway system borders the coastline (Mao
       and Rutherford, 2018a). Thus, we identified RVs by activity frequency distribution based on the navigation trajectories for
       each vessel. By defining the geographic domain of D2 in Fig. 2, vessels with more than 50% of the AIS signals throughout the

whole year occurred on inland rivers are considered as RVs (Fig. 5a). This method allows the possibilities for CVs and OGVs
       sometimes travelling into the estuaries. Finally, vessels that are not identified by OGVs and RVs are regarded as CVs.

       Figure 5b shows the identification results of OGVs, CVs and RVs, taking the year 2016 as an example. It is clear that the
       OGVs navigate between major coastal ports of China and other countries, with a few proportions entering the Yangtze River.

CVs operating around the coastal seas of China, seldom contacting with other countries. RVs mostly mainly sail on the Yangtze


River and Pearl River systems, with a small proportion wandering in coastal seas. These results of spatial distribution of OGVs, CVs, and RVs indicate that the identification method is basically satisfactory.

### 2.4.3 Ship emission control policy

In recent years, a series of policy documents have been issued to control the air pollution from ships, among which the most
effective measure is the establishment and implement of DECA (MOT, 2015, 2018). China's DECA policy were put into effect step by step from 2016 to 2019. Figure 6 summarizes the evolutionary of DECA including the control area and fuel standards, as well as their comparison with international ECA. Before the global sulphur cap taking effect in 2020, the heavy fuel oil (HFO) with sulphur content as high as 3.5% has long been used in ships all over the world. In 2015, China initially established three DECAs along the coastline (DECA 1.0), covering the most densely distributed area of ports, with gradual mandates for
ships to use low sulphur fuel (LSF) with sulphur content <0.5% m/m from core ports to the whole regions and from berthing to all operating modes, in order to reduce $SO_2$ and PM emissions. In 2018, an upgraded DECA 2.0 was proposed to expand the region to cover the entire coastline (within 12 Nm from the Chinese mainland's territorial sea baseline, Fig. 2) in which ships are required to use LSF regardless of the operating status. In addition to fuel requirement, the DECA 2.0 policy also defined the control requirement of $NO_X$ emissions from ships that diesel engine above 130 kW built or modified on or after
March 1, 2015 shall meet the Tier II $NO_X$ emission limits of revised MARPOL Annex VI rules, in line with the international ships under the control requirement of IMO.

Despite the mandatory implementation time of DECA, some regions where economic conditions permit were encouraged to experiment in advance. In order to timely feedback the effect of policies, a broad investigation of the actual performance of
DECA was conducted, including both the coastal seas and inland rivers in 2016-2019 (Supplement Table S1). Before the mandatory date of January 1st 2017, core ports in YRD and Shenzhen port pioneered the DECA 1.0 policy nine months and three months earlier, respectively. Core ports in YRD are supposed to implement the DECA 2.0 policy three months before fully coming into effect in January 1st 2019. Meanwhile, RVs are required to use the general diesel fuel (GDO) with much lower sulphur content, gradually iterating from 350 ppm to 10 ppm, finally keeping pace with the China V standard of on-road
diesel fuel in 2018.

Based on the above investigation, a policy-abutted modification module was developed in SEIM 2.0 to incorporate the actual implement of ship emission control policies with daily scale in China. At each AIS signal point, according to the geographic location, signal time and the operating mode, the vessel will be dynamically judged whether it enters the scope of DECAs at
that time and select the required fuel type and sulphur content. Then, a fuel correction factor (FCF), resulted from the quotient of the emission factors of the switched fuel and original fuel, will be further multiplied in the emission calculation formula. In this way, a high spatial and temporal resolution ship emission inventory in line with actual implementation condition of control policy will be finally developed.



### 2.4 Simulation scenario setting

To comprehensively investigate the effects of gradually implemented DECA polices under the condition of growing waterway transport demand, we designed another scenario (No-DECA scenario) in SEIM v2.0, as listed in Table 2. Compared to the base scenario embedded with actual DECA policy described in section 2.4.3, the No-DECA scenario was designed to assume vessels observed in AIS data of target year not to implement the DECA policy, namely, to simulate the ship emission of China's inland waters and the 200 Nm zone supposing all active vessels continued to use fuels with sulphur content at pre-

DECA level. By comparing the emission result from Base scenario and No-DECA scenario, the objective emission reduction effect of gradually implemented DECA policies could be vividly illustrated.

## 3 Results and discussion

### 3.1 Overall

### 3.1.1 Annual ship activities and emissions

With the development of China's waterway transport, seaborne trade has been increasing through 2016-2019. As illustrated in Fig. 7a, Chinese ports' total passenger turnover, cargo turnover and cargo throughput remained stable rise and added by 10.9%, 6.8% and 17.4% in 2019 compared to 2016, respectively. Growing water transport demand stimulated the increase of ship activities and improvement of fleet loading capacities (Table 1), which coincided with gradually implemented DECA policy and upgraded vessel engine standard, resulting in different interannual trends in ship emissions for different pollutants.


Figure 7b&c shows the annual ship emission of $SO_2$ and $NO_X$ in China's inland waters and the 200 Nm zone from 2013 to 2019. Before the enforcement of DECA policy, ship emissions of $SO_2$, $NO_X$, PM and HC in 2016 were estimated to be $1.8 \times 10^6$, $2.5 \times 10^6$, $2.3 \times 10^5$ and $1.1 \times 10^5$ Mg/year, respectively. The emission results are generally higher than other AIS-based ship emission inventories of China in recent years (Supplementary Table S2) (Chen et al., 2017; Li et al., 2018; Fu et al., 2017;

Huang et al., 2019). On the one hand, our study established a larger ship activity database based on global AIS data (~30 billion signals per year), and the incorporation of GFW database also improved the recognition of ships, especially CVs and RVs in China. On the other hand, the annual increase of ship activity driven by maritime trade could also contributed to ship emission growth. Among all vessels, OGVs composed the largest part in ship emission, with a proportion of 70.4% regarding $SO_2$ and 59.7% regarding $NO_X$ in 2016. CVs ranked after OGVs, with 29.4% contribution to $SO_2$ emission and 27.1% to $NO_X$ emission;

while RVs' composition was relatively small, accounting for 13.2% for $NO_X$ and <1% for $SO_2$. The contribution of RVs to $SO_2$ emissions was much lower than $NO_X$, as RVs were considered to use GDOs with significantly lower sulphur content than HFOs. In addition, as we identified RVs based on spatial frequency distribution of ship trajectories in AIS, which allows RVs sometimes operating in coastal waters, the identified vessels of RVs as well as emissions might be higher than that in Li et al., (2018).






From 2016 to 2018, during the DECA implement, annual ship emission of $SO_2$ around China increased by 1.6% and 3.8%, respectively; while it dropped by 33.3% in 2019 compared to 2018 owing to extended control area and more stringent requirement. Moreover, ship $SO_2$ emissions in 2019 was even 2.8% lower than that in 2013 (Fu et al., 2017), indicating the benefit of more stringent DECA 2.0 policy. Through the four years, ship emissions of $SO_2$ and PM has decreased by 29.6% 290 and 26.4%, respectively (Supplementary Table S2). In terms of $NO_X$, however, emissions continuously increased year by year, with a total increase of 13.0% from 2016 to 2019; while emissions of other pollutants also showed a gradual increase trend (Supplementary Table S2). Therefore, the ship DECA policy has a significant impact on reducing $SO_2$ and PM emission but current vessel engine emission standard only have limited influence on controlling $NO_X$ emission. In addition, although the DECA 1.0 policy has been implemented since 2017, it was not until 2019 that significant emission reduction was achieved.

**3.1.2 Contribution by flag state**

Compared to the global ship emissions estimated by *Forth International Maritime Organization (IMO) greenhouse gas (GHG) Study* (IMO, 2020), it is striking that OGVs in the 200 Nm zone of China contributed to 9.7 ~ 14.3% of global OGV emission (Supplement Table S3), despite only <1% of the world's sea area. However, we found that a considerable proportion of OGV emissions occurred in the 200 Nm zone was derived from vessels registered in other countries. Figure 8 shows the flag state 300 composition of OGV emissions operating in the 200 Nm zone, taking $SO_2$ and $NO_X$ as examples. Based on a four-year average, it turned out that OGVs registered in Mainland China, Hong Kong and Taiwan together contributed to only 31.9% and 33.4% of ship $SO_2$ and $NO_X$ emissions, respectively, in the target region. Vessels registered in Panama were the second largest OGV emission contributor besides Mainland China, holding a proportion of ~18.3%. Other major contributors of ship emissions also included Liberia (~10.0%), Marshall Islands (8.0%) and Singapore (7.7%). From the perspective of interannual change, 305 the contribution of Mainland China was raising over the four years, especially for $NO_X$, which increased from 16.3% in 2016 to 21.0% in 2019; while the second largest contributor, Panama, had declined from 19.3% to 14.5% in the same period. With the gradual effectiveness of the DECA policy in China, it is equally important to pay attention to emissions from foreign-registered ships.

**3.2 Four-year consecutive daily emission**

**3.2.1 Emission composition variation**

On a more refined time scale, we investigated the 5-day moving average ship $SO_2$ and $NO_X$ emissions on a daily basis for the inland rivers and 200 Nm zone of China from 2016 to 2019, as shown in Fig. 9. It is evident that ship emission of $SO_2$ was seasonally growing in 2016-2018, until a sharp drop on $1^{st}$ January, 2019 due to the implementation of stringent control DECA 2.0 policy. The maximum daily ship emission intensity of $SO_2$ reached $6.4\times10^3$ Mg/day on September $22^{nd}$, 2018, 2.9 times of 315 the lowest point, $2.2\times10^3$ Mg/day on January $1^{st}$, 2019; while the daily discrepancy of ship $NO_X$ emission intensity also reached





3.0 times throughout the four years. The monthly variation of ship emissions for most vessel types was generally constant except a temporary decrease during Spring Festival in February (Fig. 9a). However, fishing ships showed significant seasonal variations, which declined annually in summer and return in autumn due to fishing ban in China. This has also been demonstrated by other studies (Chen et al., 2017;Fu et al., 2017).


Figure 9 also exhibits the emission structure of $SO_2$ composed by vessel type and fuel type, and $NO_X$ composed by building year and DWT. The full composition of emission contribution for all pollutants from different aspects are summarized in Supplement Table S4. Containers had been accounted for the largest part and the contribution had been increasing through the four years, e.g., from 31.7% in 2016 to 42.9% in 2019 for $SO_2$ (Fig. 9a). Although containers accounted for only 3.5% of vessel number and 4.6% of operating hour in Chinese waters, their relatively higher engine power contributed to significant emission intensities compared to other ships of the same size, such as bulk carriers (Fig. 3). The HFO contributed to the majority of ship $SO_2$ emissions due to its high content of sulphur, part of which, however, was gradually being substituted by marine gas oil (MGO) with the implement of DECA policy (Fig. 9b). In 2019, the MGO had accounted for 15.4% of the ship $SO_2$ emission and 38.9% of $NO_X$ emission (Supplement Table S4). In terms of vessel build year, ships built after 2016 made an increasing contribution in annual NOx emission, reaching 10.6% in 2019 (Fig. 9c). Even though Tier II engine standard had been applied to domestic ships built after 2016, ship $NO_X$ emissions were not found to decrease as the emission standard of Tier II only has minor improvement compared to Tier I. In addition, we also found that ships with larger DWT have a growing proportion in vessel fleet as well as emission contribution (Table 1 and Fig. 9d), indicating the developing trend of ship upsizing in the past few years. However, even though the newly-built, large-scale ships as well as ships using clean fuel oil are all taking an increasingly large part in emission structure, the updating iteration speed of fleet is not enough to reverse the rising trend of $NO_X$ emission.

**3.2.2 Emission variation of major ports**

As is was step by step that the DECA policy was implemented in different ports in China, we extracted the 5-day moving average ship $SO_2$ emission of major ports in BRA, YRD and PRD to track the consecutive emission changes throughout the four years, as shown in Fig. 10. In the initial stage, restriction on fuels with no more that 0.5% sulphur content was only imposed on ships at berth for core ports in these three crucial port clusters (Fig. 2 and Table S1). Before the mandatory date of January $1^{st}$ 2017, core ports in YRD and Shenzhen port pioneered the implement nine months and three months earlier, respectively, which significantly showed a decrease in ship $SO_2$ emissions beginning from April $1^{st}$ and October $1^{st}$ in 2016, respectively. For other core ports in BRA and PRD, a noticeable decline could be observed on schedule on January $1^{st}$ 2017. However, emission of ships at berth took a relatively smaller percentage (7.5% ~ 13.7%) in the 200 Nm zone according to our results (Supplement Table S4), thus the emission reduction was rather conservative inside the DECA 1.0 region in 2018, even though the requirement was popularized to all ports. On the contrary, owing to intensified ship activities, ship $SO_2$ emissions for some ports even largely increased, such as Ningbo-Zhoushan Port and Shenzhen Port, increased by 19.4% and 11.4% in



2018 compared to 2017. Fortunately, in 2019, when most rigorous DECA 2.0 policy was implemented, it is clearly illustrated
in Fig. 10 that all ports' SO$_2$ emissions were dramatically reduced. Core ports in YRD were supposed to implement the DECA
2.0 policy three months before fully coming into effect. Notably, those pilots witnessed an earlier decline in SO$_2$ emission,
which also proved the timely and flexible response of SEIM 2.0 model to the changeable DECA policy.

In addition to policy-driven emission changes, different ports showed distinct monthly emission variations highly related to
their geographical location and ocean resources. For example, ship emissions in YRD region had a low point in July as their
activities were influenced by typhoon particularly in YRD (Weng et al., 2020); ship emissions in PRD region were higher in
in spring and summer since wind direction were more advantageous for ship activity in spring and summer (Chen et al., 2019b).
Besides, ship emissions in Ningbo-Zhoushan Port, Tianjin Port and Shenzhen appeared to be larger in spring and autumn,
probably owing to the large-scale fishing ship operation (Chen et al., 2016; Yin et al., 2017). The above port-based emissions
fully presented the daily ship emission variations for a long period from 2016 to 2019, which may also provide useful data
references for port observation experiments.

### 3.3 Spatial distribution change

### 3.3.1 Evaluation of route restoration

Since the shipping route restoration module was developed in SEIM v2.0 to solve the problem of AIS discontinuity, the spatial
distribution of ship emission after route restoration was evaluated, as shown in Fig. 11. Direct interpolations for AIS signals
along the loxodrome would lead to part of emissions distributing on unrealistic routes, e.g., crossing the land areas, which
could be even as long as connecting the South China Sea and the Bohai Sea (Fig. 11a). By using the route restoration method,
the ship's navigation trajectory and emissions can be restored to more realistic shipping routes, thus reducing the deviation of
the spatial distribution of emissions (Fig. 11b). Statistically, 15.3% of NO$_X$ emissions and 7.5% of SO$_2$ emissions were spatially
corrected in the study area. More improvements were obtained around Taiwan island, the Korean Peninsula and the Philippine
Islands, probably due to worse accessibility of high quality shore-based AIS signal. The misallocation of emission in China's
land areas resulted in NO$_X$ underestimate of up to 2 ~ 4 Mg/grid in the downstream of Yangtze River and Pearl River, and the
misallocation of emissions in water regions is more notable on shipping routes farther from the coast. This spatial improvement
of ship emissions with the route restoration method would be expected to reduce uncertainties in the air quality model.

### 3.3.2 Spatial change of ship emissions

Figure 12 presents the spatial change of SO$_2$ and NO$_X$ emission in 2019 compared to 2016 form ships within different coastal
region defined in Fig. 2. Remarkably, within 12 Nm, which approximately equates to the scope of DECA 2.0 in 2019, SO$_2$
emission decreased by 78.8% ($7.2 \times 10^5$ Mg/year) compared to 2016. Despite the year-by-year growth of seaborne trade, DECA
policy effectively reduced ship-emitted SO$_2$ overall and especially beneficial to coastal cities. On the other hand, however, we



discovered that $SO_2$ emission increased by 41.5% ($1.3 \times 10^5$ Mg/year) in areas between 12-50 Nm from the baseline, especially along the 12 Nm boundary. The proportions of ship $SO_2$ emission from 12-50 Nm rose from 17.5% in 2016 to 35.3% in 2019, becoming the major spatial contributor in 2019. Emission of PM exhibited the similar pattern (Supplement Fig. S2a). This peculiar phenomenon implies the fact that some ships possibly made a detour to evade switching clean fuel oil, which could also be demonstrated by the larger growth rate in cargo turnover than throughput (Fig. 6a).


Figure 12b shows that $NO_X$ emission from ships occurred within 12 Nm of the baseline were continuously increasing from 2016 to 2018, until it declined by 5.0% ($6.4 \times 10^4$ Mg/year) in 2019 compared to the last year. Meanwhile, $NO_X$ emissions occurred in areas between 12-50 Nm also turned to show a higher annual increase rate in 2019 (21.4%) than previous two years (7.4% ~ 8.2%). Such phenomenon once again proves the possibility of ship detour. Other species generally showed the

similar emission pattern as $NO_X$ (eg., HC in Supplement Fig. S2b). In sum, DECA 2.0 policy has a positive effect on ships' $SO_2$ and PM emissions control as a whole and especially for coastal areas. However, a number of ships detoured outside the scope of DECA 2.0 to save the cost on more expensive clean fuel oil, which further elongated the sailing distance and thus aggrandized emissions in farther maritime areas. This reminds us to pay attention on additional environmental effect brought by detouring ships during the continuous implementation of DECA 2.0 policy.

**3.3.3 Spatial changes of OGVs, CVs and RVs emissions**

Interannual spatial change of OGVs, CVs and RVs were further compared for ship emissions of $NO_X$ and $SO_2$, as shown in Fig. 13. Emission intensity of identified OGVs was apparently higher than CVs and RVs, demonstrating certain routes. The most intensive near-sea routes included China-Korea, China Mainland-Taiwan, the North Atlantic Route, Asia-Europe Route and routes between busy ports of China, such as main ports in BRA, YRD and PRD (Fig. 13a). Since the main shipping routes

are rather close to the land, OGVs within 12 Nm of the baseline make up for approximately 38% and 32% of total OGV emissions for $NO_X$ and $SO_2$. From 2016 to 2019, OGV emissions were generally increasing in all regions, except $SO_2$ emission in 0-12 Nm showing a significant drop down due to the DECA 2.0 policy.

As for CVs, approximately 80% of $NO_X$ emission and 70% of $SO_2$ emission were annually distributed mainly within 12 Nm

of the baseline, and the proportions occurred outside 12 Nm were greatly reduced compared to OGVs. Despite intensive emission routes between coastal ports, notable emissions from CVs occurred more evenly distributed off the major routes (Fig. 13b), which were attributed to large quantities of fishing ships operating (Kroodsma et al., 2018). In the region of 0-12 Nm to the baseline, the annual $SO_2$ emission reduction ratio of CVs (81.0%) in 2019 was even higher than that of OGVs (76.9%), indicating that CVs were more affected by the DECA 2.0 policy.


Compared to OGVs and CVs, RVs have specific routes that were constrained by inland waterways, with the most intensive emission located on the Yangtze River and the Pearl River (Fig. 13c). Meanwhile, RVs also operate along Chinese coastal and



produce a considerable proportion of emissions within 12 Nm of the baseline. With the increasingly stringent national fuel oil standards for RVs (MEE, 2018), i.e., sulphur content from 350 ppm before June 30th, 2017 to current 10ppm beginning from
January 1st, 2018, $SO_2$ emissions from RVs had been reduced to a rather low level, both for inland rivers and coastal areas. However, other pollutants such as $NO_X$ emissions from RVs were still going uphill. In addition, although China has required certain categories of ships to install AIS equipment since 2010 step by step, a large part of small RVs in China have not been equipped with AIS (Zhang et al., 2017). The lack of ship activity level and highly reliable local emission factors all brings uncertainties to the emission estimation of RVs. However, air quality and human health of inland cities near waterways could
be impacted severely by RVs emissions (Wang et al., 2018). Therefore, RV emissions need to be stressed and worth further investigation.

### 3.4 Emission reduction effect of DECA policy

### 3.4.1 Monthly effect evaluation

Since the shipping activity increase and emission control collectively resulted in the past emission trend, we designed another
scenario without DECA policy to evaluate the emission reduction effect considering the annual change of ship activities. Figure 14 illustrates the monthly ship emissions of $SO_2$ for base (real) case and the No-DECA scenario, which are aggregated from inland rivers and the 200 Nm zone of China. Without DECA policy, ship emissions of $SO_2$ were estimated to increase from $1.8\times10^6$ Mg/year in 2016 to $2.1\times10^6$ Mg/year in 2019, with an annual increase rate at 4.5%. Beginning from April 2016, the prior implement of DECA 1.0 led by core ports of YRD started to see the emission reduction benefit. Since DECA 1.0, ship
$SO_2$ emissions reduced by $4.6\times10^4$, $1.1\times10^5$, and $1.4\times10^5$ Mg/year for 2016, 2017 and 2018, respectively, compared with No-DECA scenario. Emissions were cut down even more remarkably in the year of 2019 owning to the expansion of DECA 2.0, with $8.4\times10^5$ Mg $SO_2$ reduced compared to No-DECA scenario. In retrospect, although ship $SO_2$ emission had been reduced by 29.6% in 2019 compared to 2016 in base scenario, it actually achieved a larger benefit with a reduction of 39.8% compared to the same year considering the actual seaborne trade growth and ship activity increase.

### 3.4.2 Annual regional contribution

So far, the implementation of DECA policy and the effect of ship emission reduction have been focused within 12 Nm of the baseline of China's territorial sea. To further investigate the regional contribution of emission changes in different regions, we finally summarized ships' activity and emissions in BRA, YRD and PRD, respectively, from 2016 to 2019, as shown in Fig. 15. Although the annual change of $SO_2$ emission in 2017 and 2018 was not significant, i.e., decreased by 3.9% and increased
by 1.3%, respectively, during the implement of DECA 1.0; it is undeniable that the policy had indeed effectively reduced emissions as the growth of ship activities would lead to 7.9% and 17.1% increase in emissions if without the DECA 1.0 policy. Moreover, YRD and BRA played a leading role in reducing ship $SO_2$ emission in 2017 and 2018, respectively. However, further tightened DECA 2.0 policies implemented in 2019 more effectively reduced $SO_2$ emissions by 78.2%, in which YRD,





BRA and PRD contributed 30.1%, 20.2% and 16.2%, respectively, while other waters contributed the rest 26.7%. Therefore,
even the controlling area of DECA 2.0 was enlarged to 2.5 times of DECA 1.0, the dominant regions of emission reduction
were still the three major port clusters. The primary factor driving DECA 2.0 achieving a larger amount of emission reduction
is the fuel switching regulation for all operating status of ships sailing in the region rather than limiting only berthing status in
DECA 1.0.

## 4 Conclusion and policy implication

### 4.1 Conclusion

The DECA policy had effectively reduced $SO_2$ and PM emissions from ships in sea areas around China from 2016 to 2019.
Although the preliminary DECA 1.0 policy targeting mainly on berthing ships only had limited effects on ship-emitted $SO_2$
and PM, the DECA 2.0 policy, tightening its limitation by putting ships in all operating status under control and expanded the
control areas from major ports to 12 Nm from the Chinese mainland's territorial sea baseline , resulted in significant emission
reduction. As a result, $SO_2$ and PM emissions from ships decreased by 29.6% and 26.4% in 2019 compared to 2016,
respectively. Considering the potential emissions brought by continuous growth of maritime trade, a more substantial benefit
was even achieved, e.g., an emission reduction of 39.8% regarding $SO_2$ in 2019 compared with the scenario without any
emission control policy. However, $NO_X$ emissions from ships increased by 13.0% throughout the four years, indicating the
limited effect of current control standard. In addition, although OGVs contributed approximated two-thirds of ship emissions
in the 200 Nm zone of China, two-thirds of them come from ships registered in other countries.

Based on a four-year consecutive daily emission analysis, it is noticeable that the ship emission structure had been gradually
changing, i.e., newly built, large ships and ships using clean fuel oil were taking an increasingly large proportion in emission
structure. Containers and bulk carriers were still the dominant vessel type in ship emission composition. On a local scale, ship
emissions in various ports exhibited different patterns in terms of daily variation. For example, ports in YRD were likely to
encounter typhoon in July and fishing ships were particularly abundant in BRA. Relevant findings may help provide useful
data references for port observation experiments and local policy making.

The interannual spatial change of ship emissions also showed new characteristics. Through contrasting ship emissions within
different distance from Chinese coastal baseline, we discovered that in 2019, a number of ships detoured outside the scope of
DECA 2.0 probably to save the cost on more expensive low sulphur oil, which further elongated the sailing distance and thus
aggrandized emissions in farther maritime areas. This reminds us to pay attention on additional environmental effect brought
by detouring ships during the continuous implementation of DECA 2.0 policy. In addition, the route restoration method
developed in SEIM v2.0 effectively restored ship's navigation trajectory and emissions to more realistic shipping routes, thus

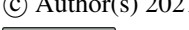



reducing the deviation of the spatial distribution of emissions and could be expected to reduce uncertainties in the air quality model.

## 4.2 Policy implication

Compared to the increasingly strict emission control policies of land-based sources and improving air quality in China, policies and regulations for the prevention and control of ship emissions would be more urgent to facilitate China's air quality to

achieve the annual $PM_{2.5}$ concentration standard of World Health Organization (WHO) Air Quality Guidelines (Wang et al., 2020; Zhang et al., 2019a). Although the current emission policy has achieved significant control effect on $SO_2$ and PM emission, under the global low sulphur oil demand, China still needs to further apply for international ECA to enlarge the control area and strengthen the requirements for fuel quality. In order to make a comprehensive evaluation and in-depth improvement of the policy, attention is also needed during the design process of ECA scheme, such as the corresponding

impact of ship detour and further expand DECA 2.0 so as to enlarge the reduction effects within 200 Nm zone. Meanwhile, the international cooperation is also urgently called for to jointly control ship emissions due to ships' strong spatial mobility and the complexity of registration and operation. With the gradual cleaning of marine fuel and the obsolescence of HFO, ship emissions of $SO_2$ and PM will be effectively mitigated in the near future. However, ship $NO_X$ emissions are still expected to increase until the gradual elimination of old ships and iteration of more stringent Tier III standard for newly built ships. Other

related factors, such as engine type, $NO_X$ post-treatment technology etc. should be taken into consideration in the future. For local decision makers, it is also important to make clear the local ship emission structure and meteorological conditions in order to conduct effective measures.

## Data availability

The AIS data and STSD are restricted to the third party and used under license for the current study.

## Code availability

Python codes used during the current study are available from the corresponding author on reasonable request.

## Acknowledgements

This work is supported by the National Natural Science Foundation of China (grant nos. 42061130213 and 41822505). H.L. is supported by the Royal Society of UK through Newton Advanced Fellowship (NAF\R1\201166).

## Author contributions

XW and WY contributed equally. XW and WY designed the research and wrote the manuscript. LZ and DF provided multiple analytical perspective on this research. ZS and XH helped collect and clean the ship data. LH provided guidance on the research and revised the paper. All authors contribute to the discussion and revision.



**Competing interests**

The authors declare no competing interests.

**Additional information**

Supplementary information is available for this paper at online resources.

Correspondence and requests for materials should be addressed to H.L.




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


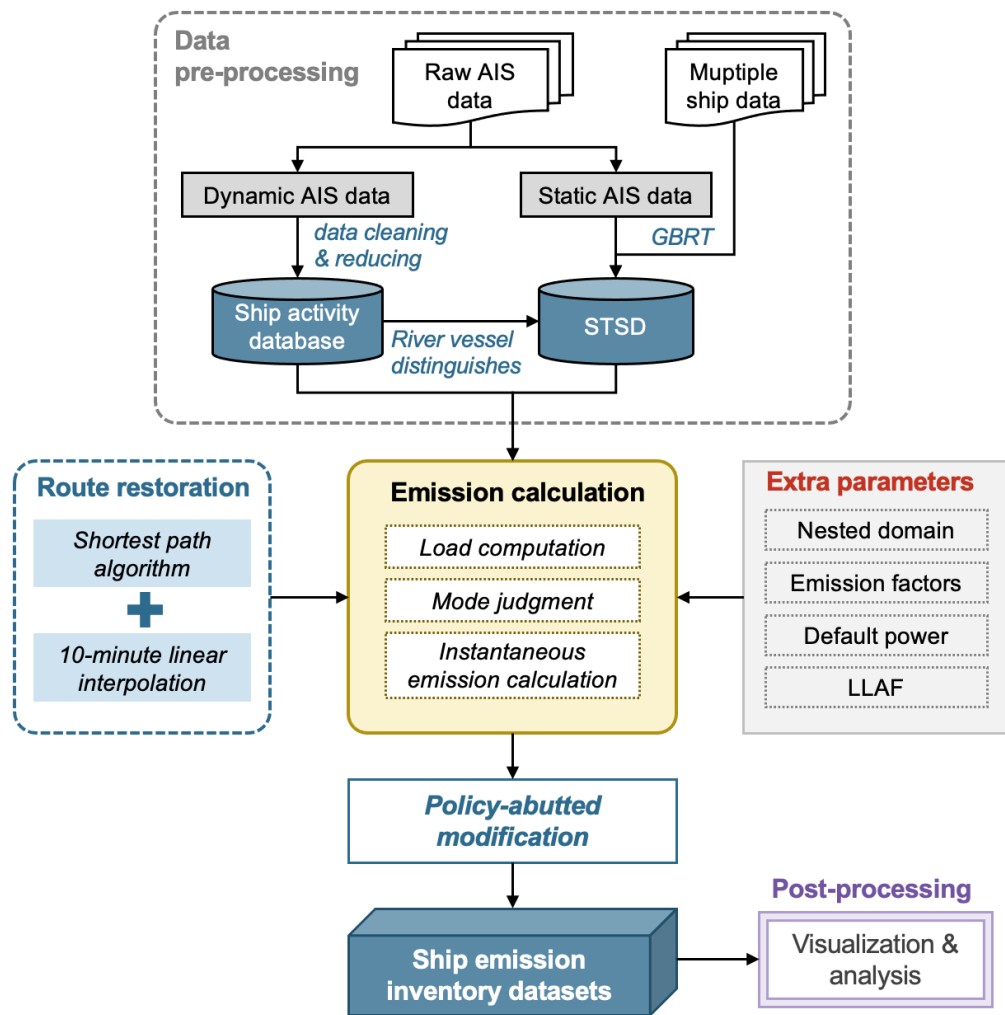

**Figure 1: Structure and flow chart of SEIM v2.0. The STSD stands for the ship technical specification database. The GBRT stands for the Gradient Boosting Regression Tree. The LLAF stands for the low load adjust factors.**


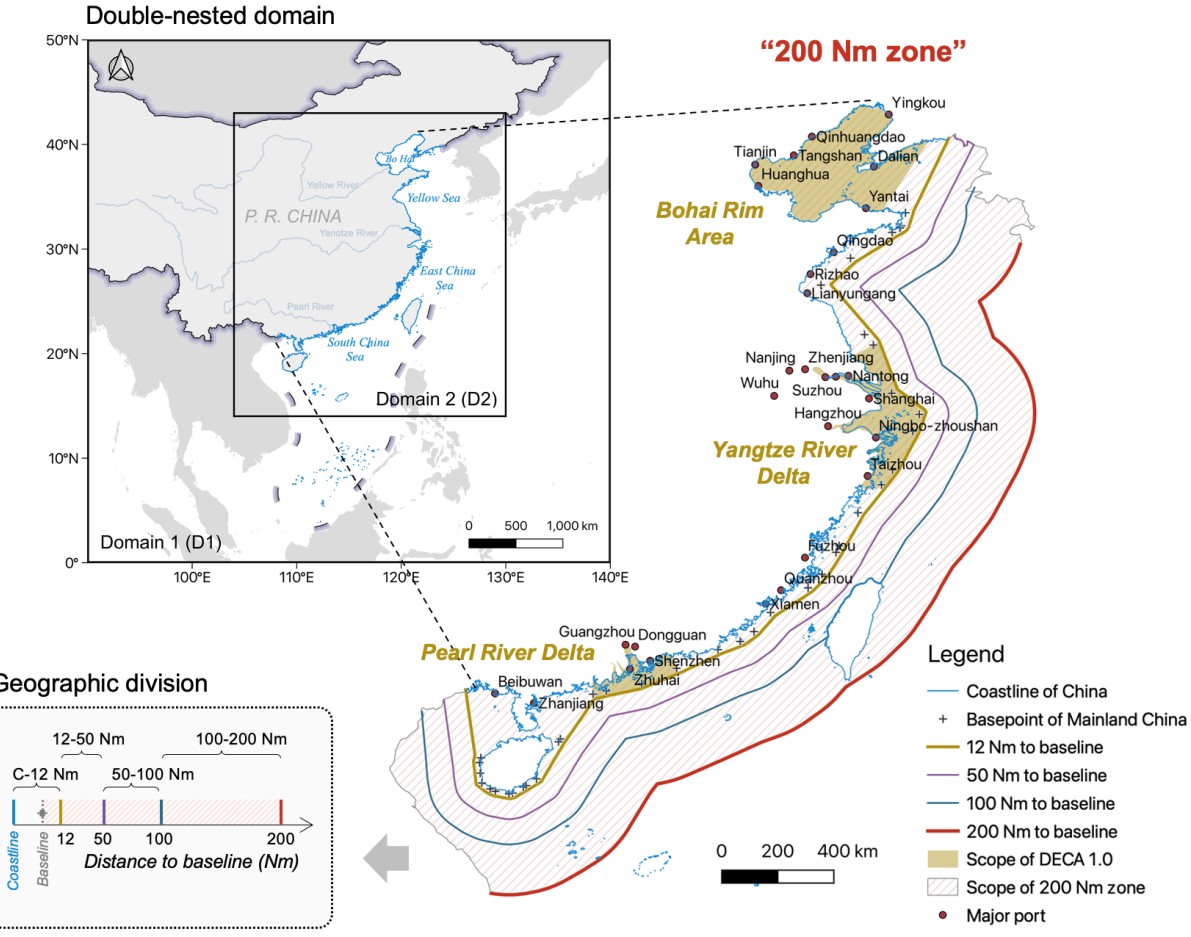

**Figure 2: Definition of the study area for ship emission estimation around China. The double-nested domain is used to filter global AIS data and reduce the boundary effect. The distances in the map all refers to the distance from the baseline of the Chinese mainland's territorial waters. The 200 Nm zone is the coastal area approximately within 200 Nm away from the baseline, which is further divided to different geographic regions according to the distance lines.**





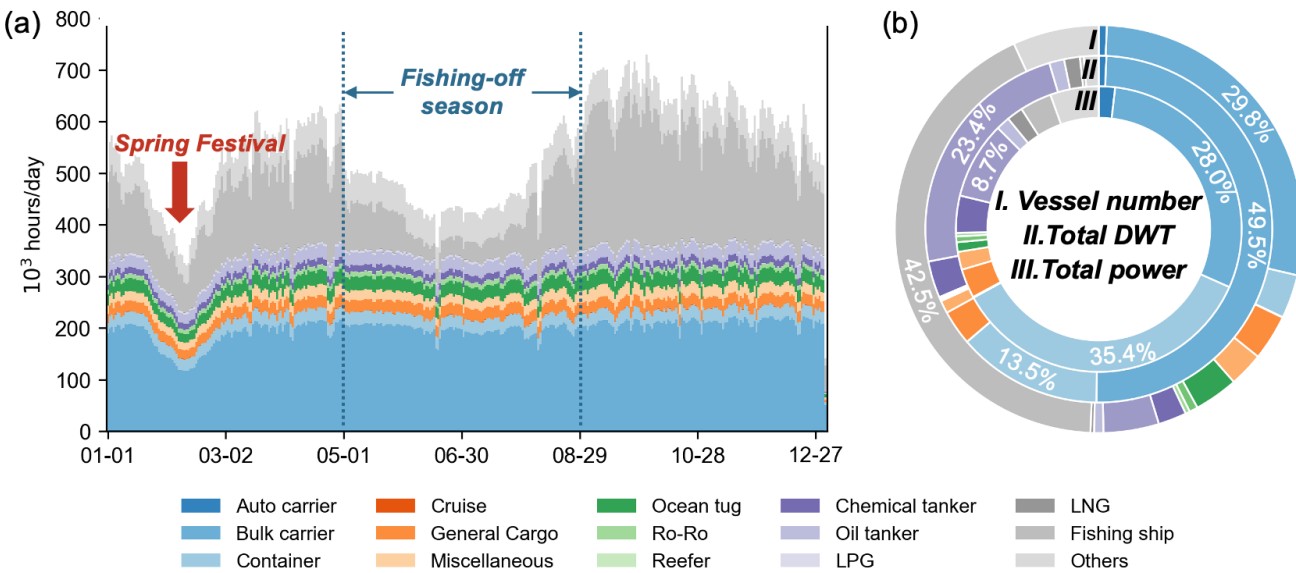

**Figure 3: Vessel dynamic and static information statistics. (a) Daily average operating hour averaged for 2016-2019. (b) Vessel fleet compositions from different aspects.**





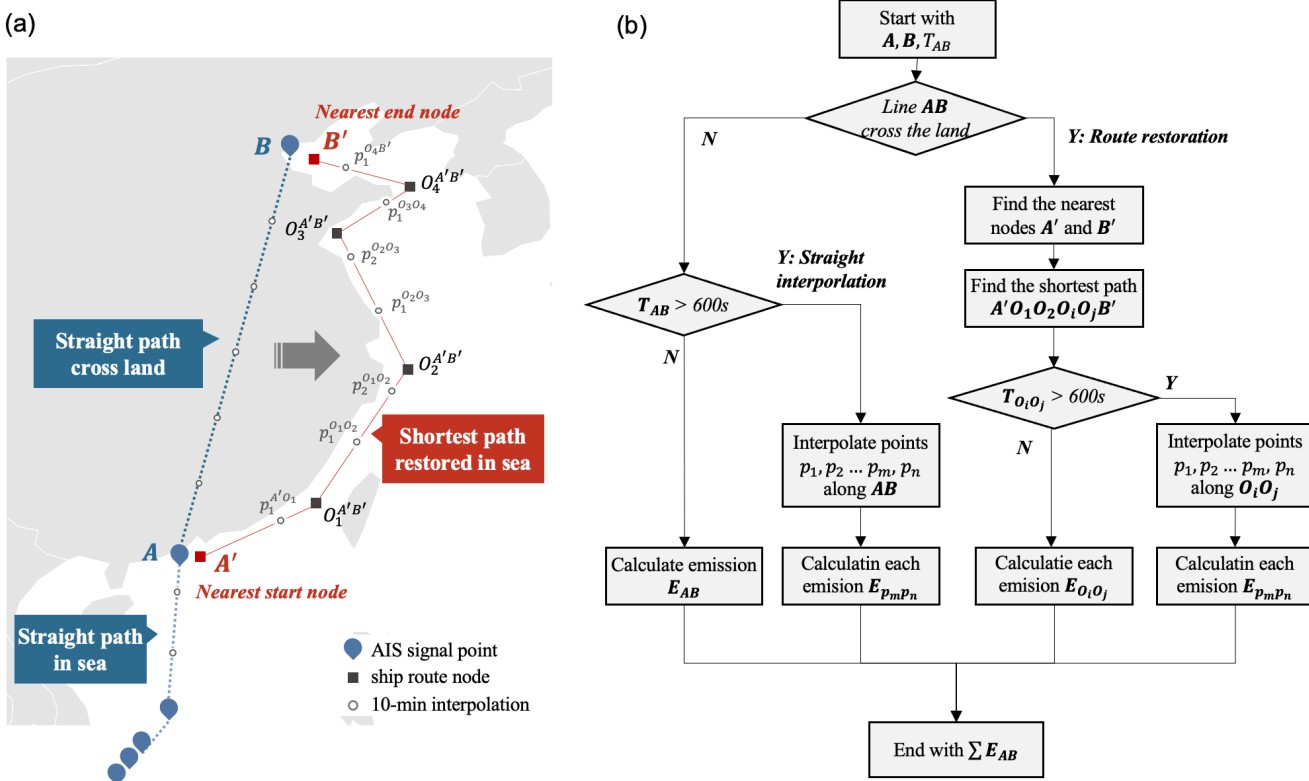

**Figure 4: Diagrammatic sketch of the ship route restoration algorithm. (a) Sketch map of the route restoration algorithm with an example of route AB. (b) Algorithm flow chart of the example of route AB.**

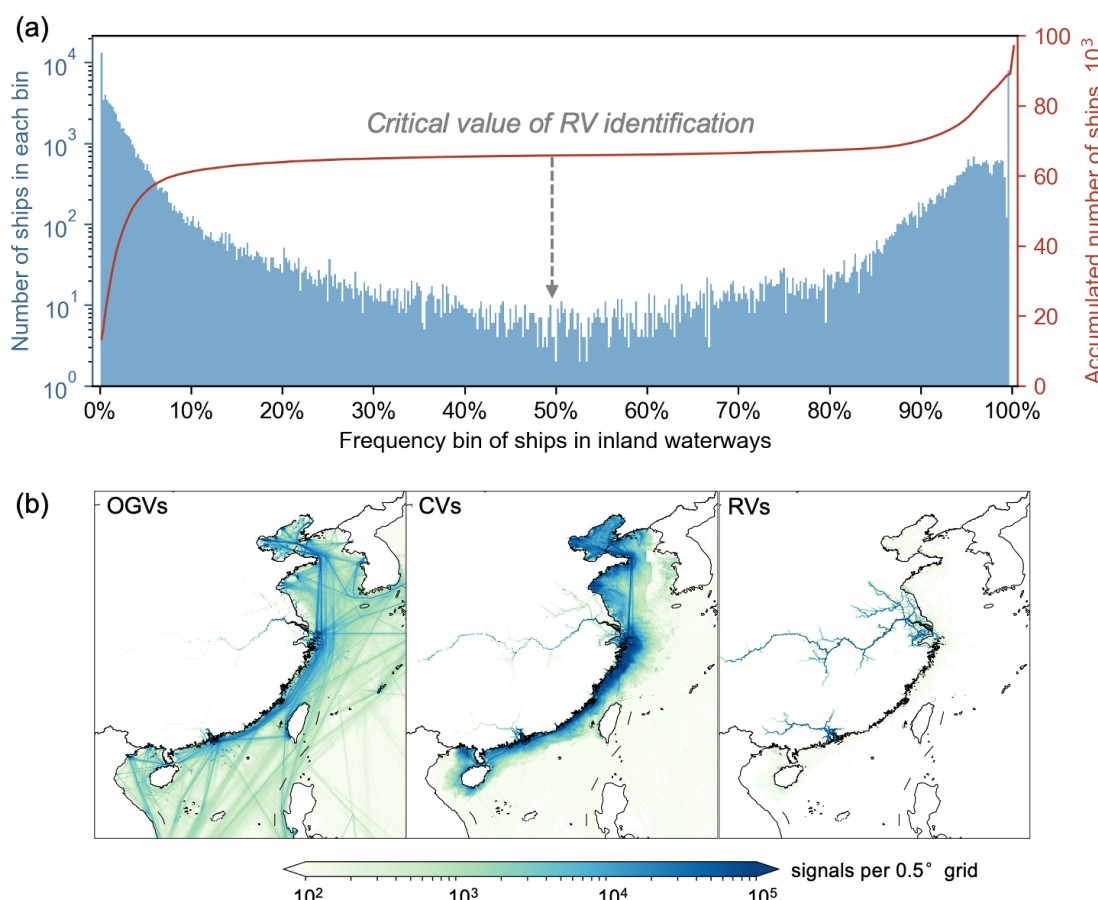

**Figure 5: Identification test and results of OGV, CV and RV. (a) Frequency test of ships in inland waterways. (b) Spatial distribution results of AIS signals of OGVs, CVs and RVs. The sample year is 2016.**



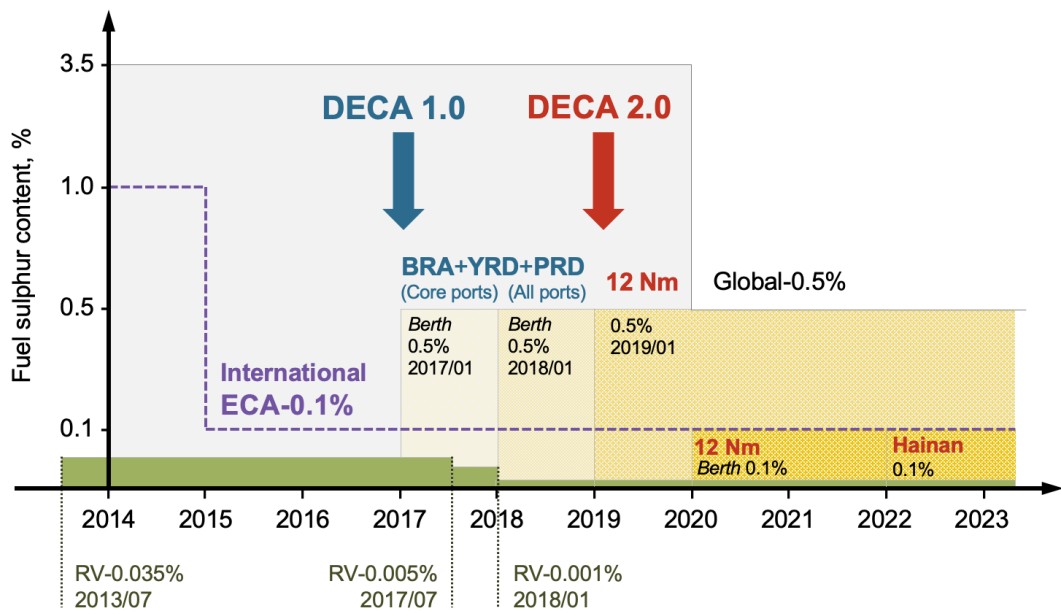

**Figure 6: Evolution of sulphur content requirements for fuels in DECAs and inland rivers in China. The percentages refer to the sulphur content of the fuel. The italics refer to the operating mode constrained by DECA policy. The yellow shadow represents the control requirements of the coastal DECA, in which darker color means more stringent requirements; while the green background represents that for river vessels.**





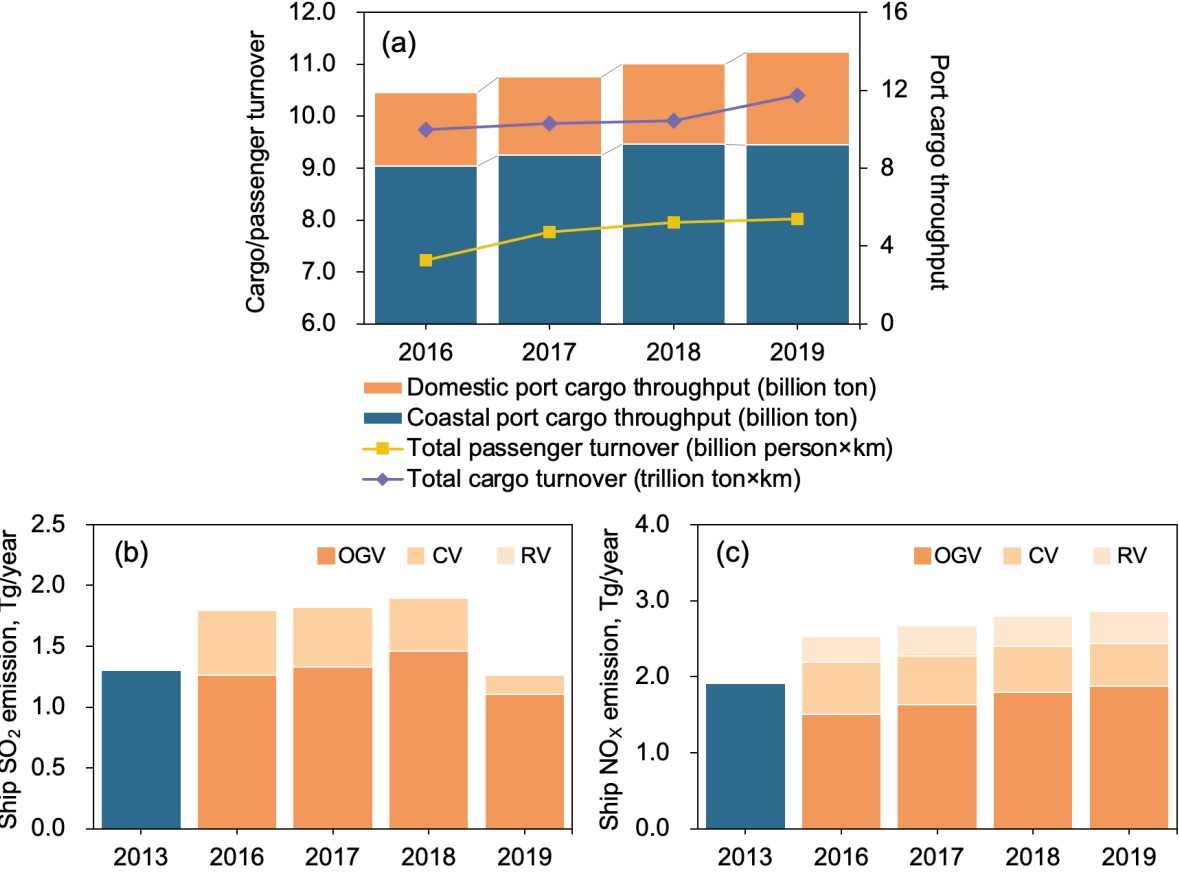

**Figure 7: Annual changes of (a) seaborne trade and ship emissions of (b) SO₂ and (c) NOₓ from 2016 to 2019. Data in (a) are collected from Chinese Statistical Yearbook (NBS, 2020). Emissions of 2013 are derived from our previous work for comparison (Fu et al., 2017).**





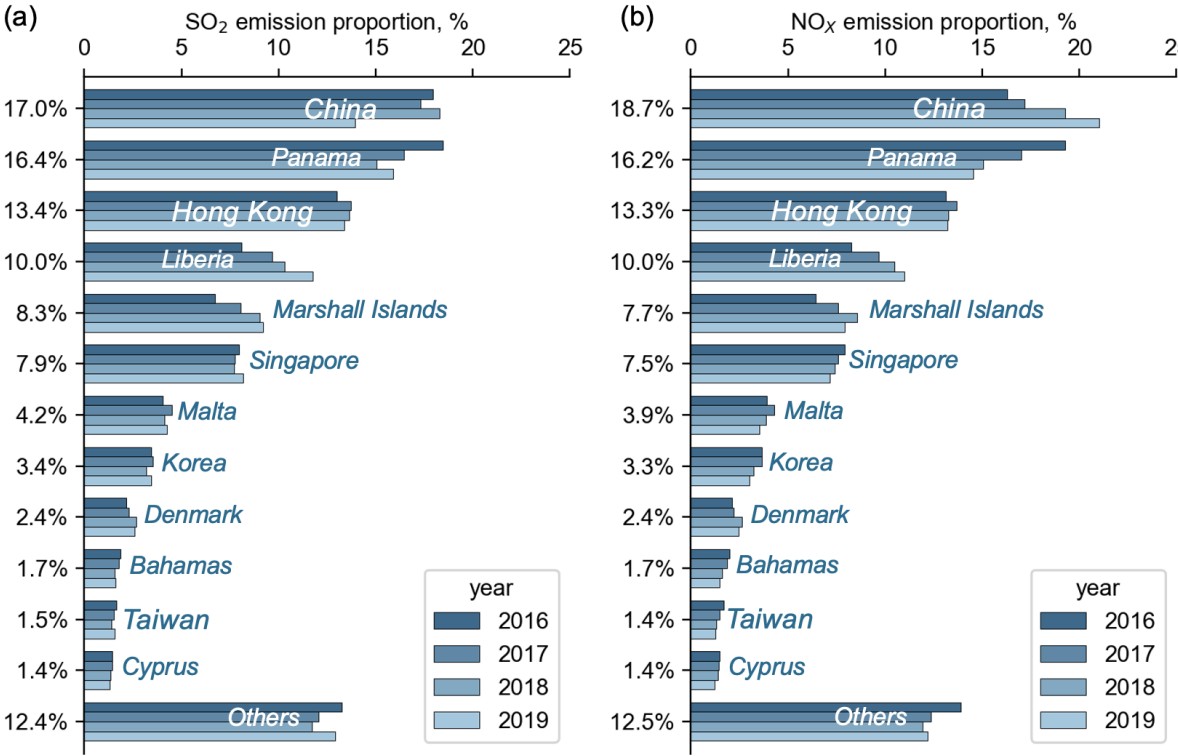

**Figure 8: Emission proportion of OGV for (a) SO₂ and (b) NOₓ from different vessel flag state in the 200 Nm zone of China. Countries/regions are arranged in descending order of four-year average proportion.**





**Figure 9: The 5-day moving average of SO₂ and NOₓ emissions from ships around China from 2016 to 2019. Ship SO₂ emission composition of (a) vessel type, (b) fuel type, and ship NOₓ emission composition of (c) vessel build period and (d) dead weight tonnage (DWT).**


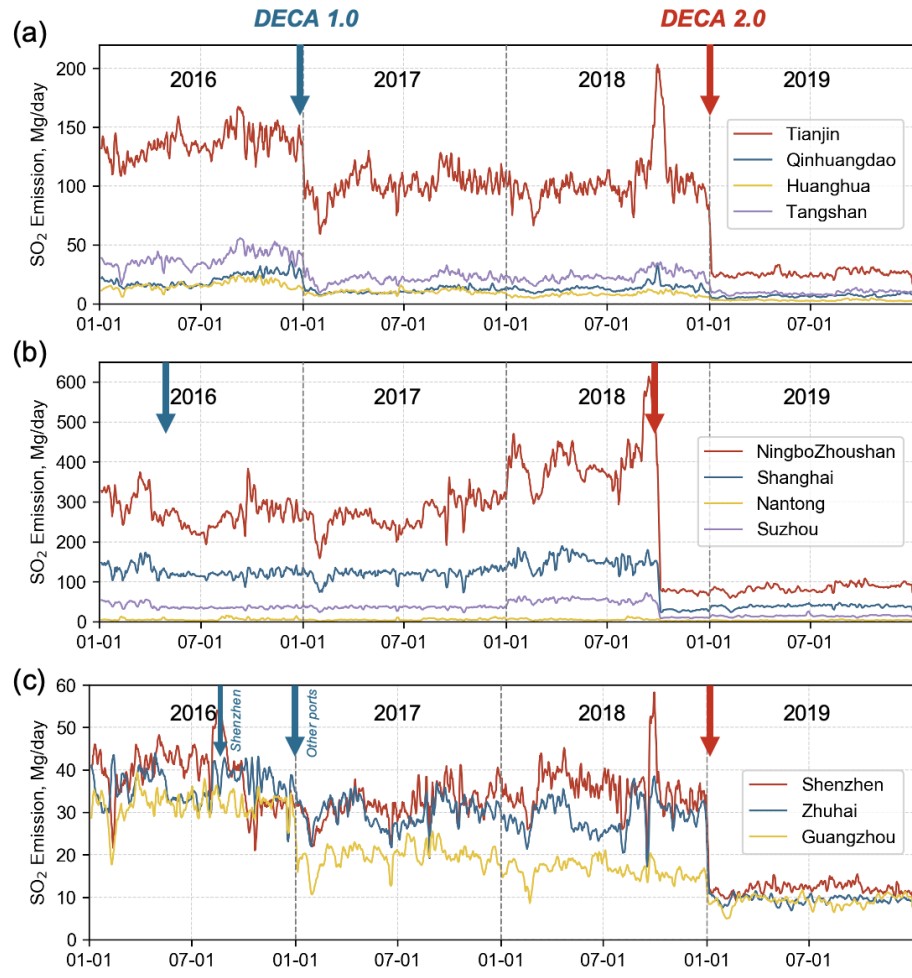

**Figure 10: The 5-day moving average of SO₂ emission from ships in major ports of China from 2016 to 2019. (a) Bohai Sea Area (BRA), (b) Yangtze River Delta (YRD), and (c) Pearl River Delta (PRD). The blue and red arrows mark the actual implementation dates of DECA 1.0 and DECA 2.0 policies, respectively.**



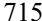


**Figure 11: Evaluation of estimating ship NO$_X$ emission in China after route restoration. (a) Emission without route restoration. (b) Emission with route restoration. (c) Spatial difference of emission (a-b). (d) Spatially change rate of emission, i.e., (a-b)/a. The selected year is 2016.**




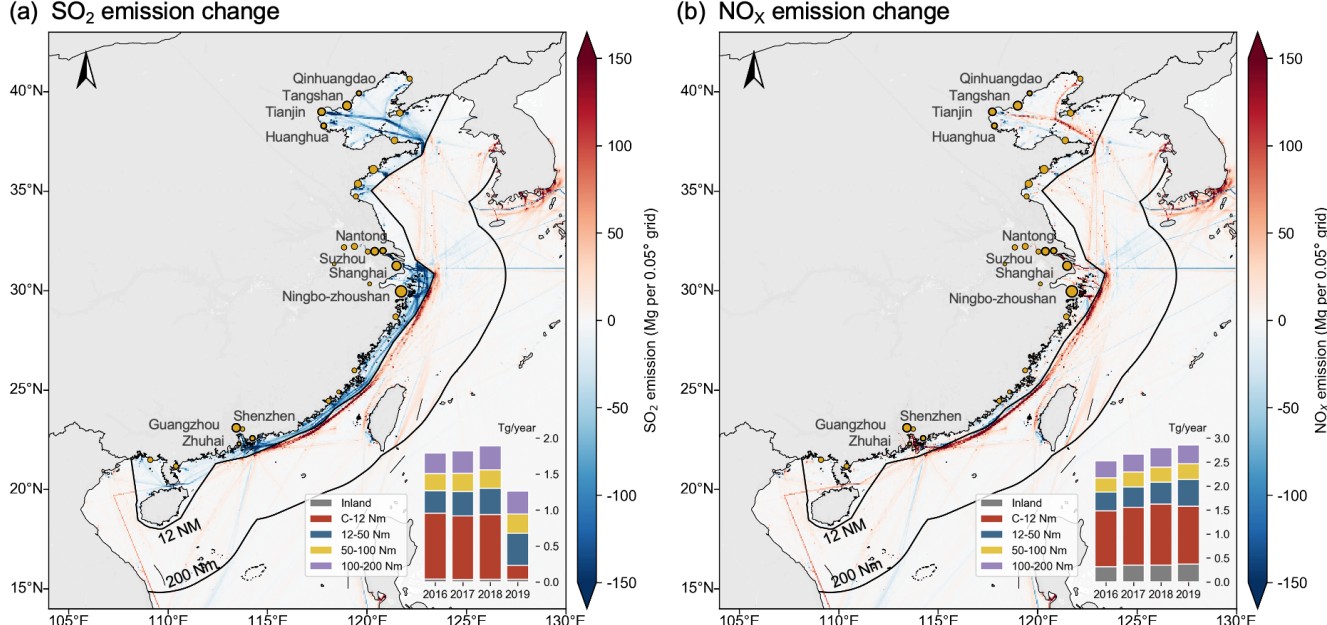

**Figure 12: Spatial distribution changes of SO₂ and NOₓ emissions from ships over China in 2019 compared to 2016. The stacked bar plots indicate the annual emission occurred at different distances off the coastline from 2016 to 2019. The "C-12 Nm" in the legend refers to the area from the coastline to 12 Nm from the baseline of the territorial sea (the same below), which is approximately equal to the scope of DECA 2.0.**




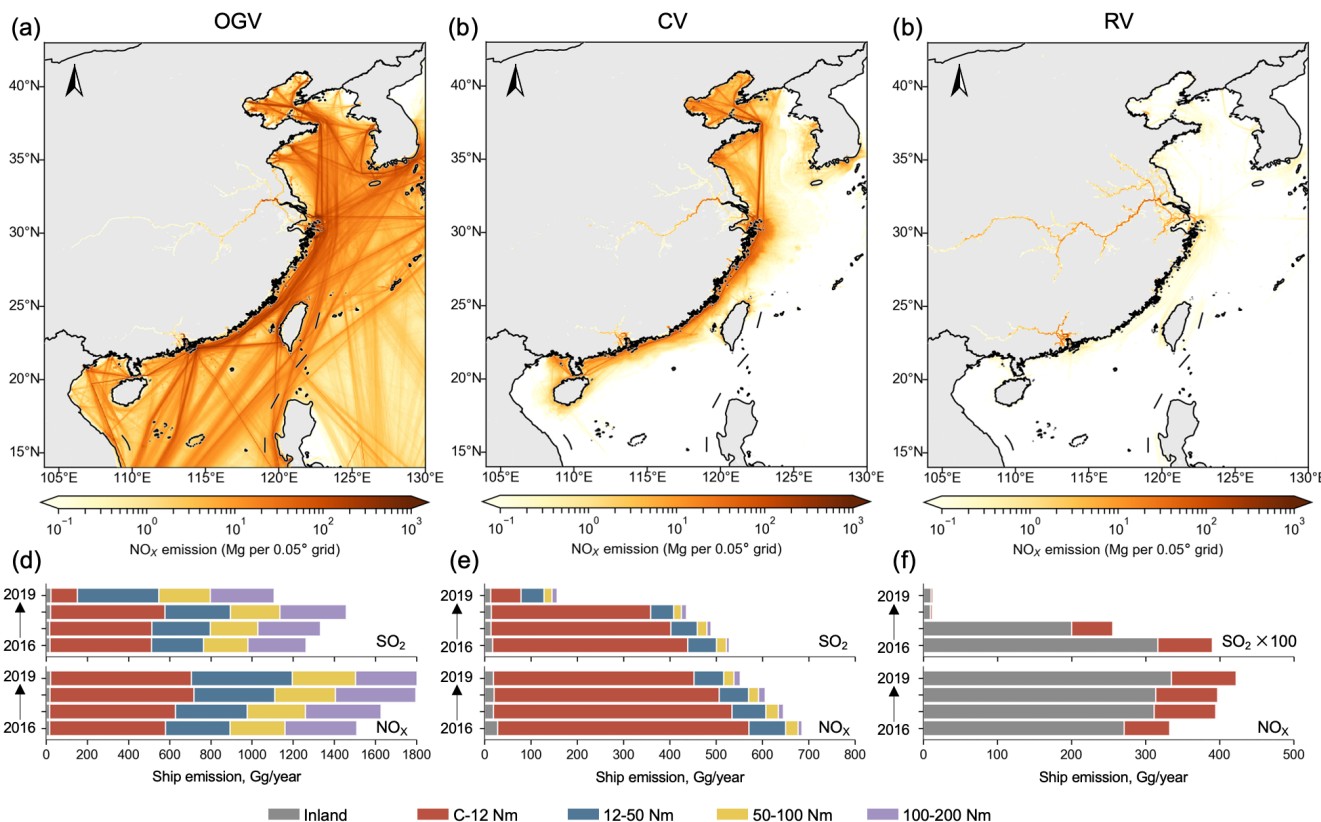

**Figure 13: Interannual spatial change of NO$_X$ and SO$_2$ emissions from ships over China from 2016 to 2019. Annual average spatial distribution comparation of NO$_X$ emission for (a) OGVs, (b) CVs and (c) RVs. Interannual variations of NO$_X$ and SO$_2$ emission in different geographic regions for (d) OGVs, (e) CVs and (f) RVs.**



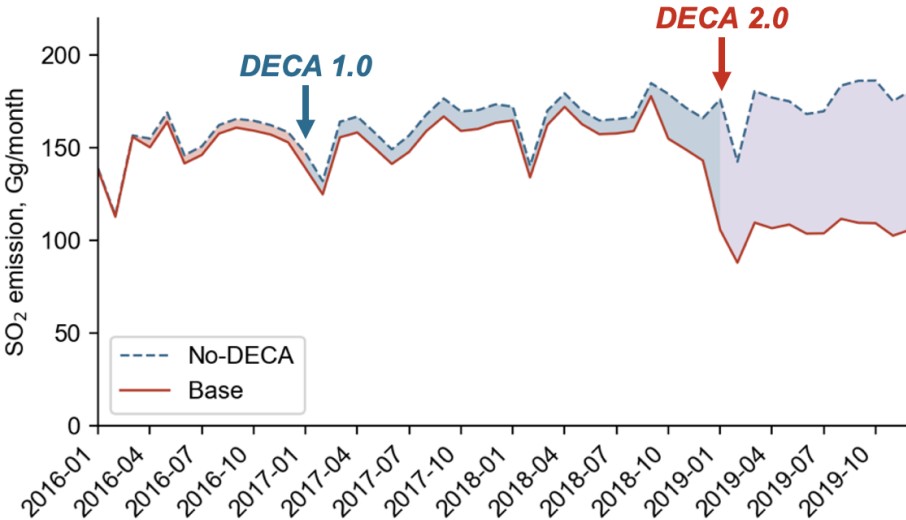


**Figure 14: Monthly variation of ship SO₂ emissions of China under Base and No-DECA scenarios in 2016-2019. The Base scenario refer to the real condition. The No-DECA scenario reflect the emission based on the real ship activities without the DECA policy.**





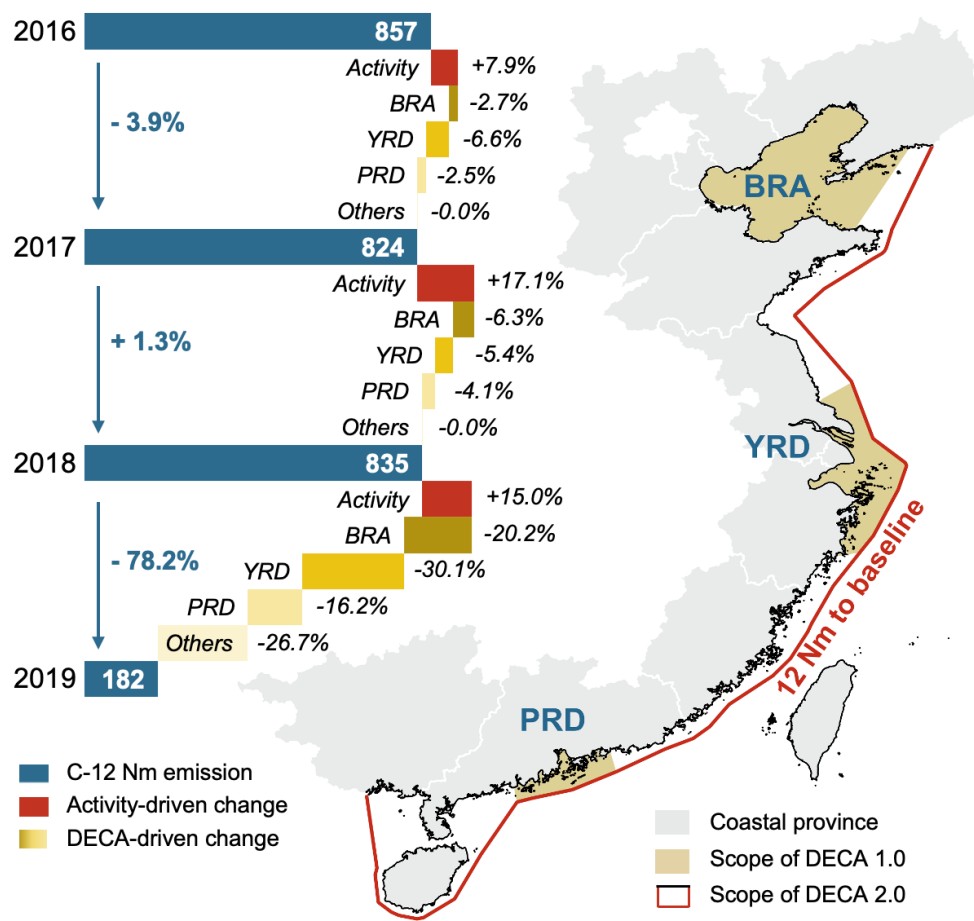


**Figure 15: Regional contributions to annual reduction SO₂ emissions from ships within within 12 Nm of the baseline of China's territorial sea. The figures inside the blue bars refer to the annual emissions, and the percentages refer to the relative change of emissions due to total ship activity change in C-12 Nm region or the DECA policies in each region.**






**Table 1: Statistics of AIS messages and active ships in China in 2016–2019.**

| Statistical items | | 2016 | 2017 | 2018 | 2019 |
|---|---|---|---|---|---|
| **Global** | Archived AIS messages ($10^9$) | 26 | 35 | 31 | 45 |
| | Active ships with unique MMSI ($10^3$) | 523 | 635 | 754 | 824 |
| **China** **(200 Nm zone)** | Number of identified ships ($10^3$) | 96 | 92 | 88 | 85 |
| | Total operating hours ($10^6$ hours) | 196 | 197 | 195 | 202 |
| | Total main engine power ($10^6$ kW) | 381 | 398 | 387 | 378 |
| | Total dead weight tonnage ($10^6$ tons) | 1340 | 1436 | 1481 | 1515 |



**Table 2: Simulation scenario setting in this study.**

| Scenario | AIS data | Coastal sea | | Inland river | |
|---|---|---|---|---|---|
| | | **Policy setting** | **Fuel setting** | **Policy setting** | **Fuel setting** |
| Base | 2016-2019 | Actual implement of DECA 1.0 and DECA 2.0 | Inside DECAs : LSF (S < 0.5% m/m) <br> Outside DECAs: No requirement | As required | 350 ppm, 50 ppm and 10 ppm chronologically |
| No-DECA | 2016-2019 | No Policy of DECA | Pre-DECA level (No requirement) | Assumed fuel | 350 ppm |