# Peer review of "Annual changes in ship emissions around China under gradually promoted control policies from 2016 to 2019"

_Atmospheric Chemistry and Physics, 2021_

## Author Comment (AC1)

**Response to Reviewers #1's Comments**

**Summary:**

This paper presents the development of an emissions inventory for ship emissions from the costal and river waters around China, based on the Ship Emission Inventory Model (SEIM), using a comprehensive set of AIS data to produce a high temporal and spatial resolution inventory. It uses the inventory, along with changes in emissions due to policy interventions as part of the domestic ship emission control (DECA) policy, to assess how ship emissions around China have changed from 2016 to 2019. Emission from ships play an important role in air pollution in China (as they do globally) and as a result studies like this are important in order to fully understand the effect. This work is novel, interesting and reasonably well written. I believe it should be published in ACP if the authors can address the following minor points.

**Response:**

Thank you very much for spending time to give us so many constructive comments. Upon learning through them, we improved our manuscript and tried our best to address all the concerns in this revision.

**General comments:**

**Q1. Description of AIS data**

I found the description of how the AIS data of number and position of ships is actually turned into emissions of air pollutant a bit lacking. What emission factors are used for this? Connected to this, the authors use the China domestic ship emission control (DECA) policy to alter emissions throughout the study period but do not quote any evidence as to how effective the policy has been or how much compliance there has been with the emission reductions. Are there any measurement studies that could be quoted to show whether ships are sticking to the DECAs?

**Response:**

Thank you for the comment.

**(1) Emission factors**

The emission calculation process within SEIM model has been introduced in our previous study (*Liu et al., 2016; Fu et al., 2017; Liu et al., 2018*) in detail, including the data collection and cleaning, calculation formula, emission factor (EF) adoption as well as default parameter setting. This article inherited the original SEIM model and upgraded it to the SEIM v2.0. To better guide readers, we added more descriptions related the calculation methods and the data in the Supplementary

Methods, including:

- a) General calculation formula in SEIM v2.0;
- b) Automatic Identification System (AIS) data;
- c) Ship technical specifications database (STSD);
- d) Emission factors (EFs) connected with the DECA policy.

Correspondingly, we also added **Fig. S1** to examine the temporal and spatial coverage of AIS data, and **Table S2** to show the emission factors applied in the SEIM v2.0. Thank you for your kind remainder.

**(2) Compliance of DECA policy**

DECA policy came into effect guaranteed by the authority of Chinese government, which endowed it with the power to restrict the fuel ship-owners uses. Several investigations, using whether field measurements or simulations, have showed that with DECA policy, not only ship fuels have been found to be cleaner, but also air pollution caused by shipping activities have been evaluated to be less in important ports alongside Chinese coast (*Liu et al., 2018; Zhang et al., 2019; Zhao et al., 2020; Zou et al., 2020*). However, there are not sufficient evidence indicating all vessels stick to DECA's regulation or the violation rate of DECA policy from year to year. This could cause uncertainties in this study. Therefore, an explanation was added in the revised manuscript to address this uncertainty.

**Revisions in Manuscript:**

- 1) Lines 108-109: The general calculation formula of the SEIM model is summarized in the Supplementary Methods.
- 2) Lines 290-297: The details about the emission factors regarding different fuel types were introduced in the Supplementary Methods. It is worth noting that as far as we know, there has not been sufficient evidence showing all vessels are sticking to DECAs or the violation rate each year. But there are studies indicating the effectiveness of DECAs in recent years (Liu et al., 2018; Zhang et al., 2019; Zhao et al., 2020; Zou et al., 2020). Not only have fuels been found to be cleaner (Zhang et al., 2019), but also air pollution caused by shipping activities has been less in important ports alongside Chinese coast (Zou et al., 2020). Guaranteed by the authority of Chinese government, we assume that the DECA policy should mostly be effective, but lack of evidence about the violation of DECAs added to uncertainties in this model.

**Revisions in Supplement:**

**General calculation formula in SEIM v2.0**

The emission calculation in this study was made for each individual vessel, with a

breakdown into three different engine types (main engine, auxiliary engine, and boiler) and four operation modes (At berth, at anchorage, maneuvering, and at sea). Transient emissions are calculated by multiplying emission factors (per unit power) by engine load ratios, with adjustment factors for fuel type and sulfur content. Total emissions are aggregated using transient emissions multiplied by time durations. The equations (1), (2) and (3) provide the emission calculation for main engine (ME) auxiliary engine (AE) and boiler in SEIM v2.0 model.

$$E^{\rm ME} = \sum_{t=1}^{n} {\rm MCR} \times {\rm EF}_{p,i,j,k}^{\rm ME} \times {\rm LF}_t \times {\rm LLAF}_p \times \Delta T_t \times 10^{-6}$$
(1)

$$E^{AE} = \sum_{t=1}^{n} P_{\nu,s,m}^{AE} \times EF_{p,i,k}^{AE} \times \Delta T_t \times 10^{-6}$$
(2)

$$E^{\text{Boiler}} = \sum_{t=1}^{n} P_{\nu,s,m}^{\text{Boiler}} \times \text{EF}_{p,i}^{\text{Boiler}} \times \Delta T_t \times 10^{-6}$$
(3)

Where MCR is the maximum continuous rated power (kW) for each vessel;  $EF_{p,i,j,k}^{ME}$  is the emission factor for fuel type *i*, engine type *j*, emission standard *k* and species *p* (g/kW·h); LFt is the load factor in time interval *t*, LLAFp is the low load adjust factor for species *p*, which is applied when the load factor is less than 20%;  $\Delta T_t$  is the time interval of the *t*-th continuous AIS signal (h); *n* is the total number of AIS signal time intervals under each category.  $P_{v,s,m}^{AE}$  and  $P_{v,s,m}^{Boiler}$ is the operating power (kw) of AEs and boiler of ship type *v* and size bin *s* (divided by dead weight tonnage) under operating mode *m* (kW);  $EF_{p,j,k}^{AE}$  is the emission factor of pollutant *p* for AEs using fuel type *i* and complying with emission standard *k* (g/kW·h);  $EF_{p,i}^{Boiler}$  is the emission factor of pollutant *p* for boilers using fuel type *i* (g/kW·h). Detailed description was provided in the Methods of our previous study (Liu et al., 2016).

**Automatic Identification System (AIS) data**

The temporal and spatial coverage of AIS data were examined to guarantee the quality of ship emission inventories. The full year AIS data including both satellite signals and territorial signals from 2016 to 2019 were used for our emission calculations in this study. Fig. S1a showed the homogeneity of the AIS signals in this study in terms of time. It is noticeable that during February (Spring Festival Holiday in China) and May to August (Fishing-off Season in China), the number of daily AIS signals is lower than average (approx. 5 million/day). Missing signals or anomalies occasionally exist, which could due to multiple factors, such as disruption to satellites, equipment maintaining, data transmission fault etc., Besides, Bad weather could be a reason for interference of signal transmission. After the

adoption of the 10-minute interpolation method, the AIS signal is expanded to about twice the original, and some periods with long intervals have been obviously supplemented. Fig. S1 (b) and (c) exhibited the change of spatial coverage of AIS signals in inland waters and coastal waters around China. The number of AIS messages transmitted per year is increasing over the span of this study's years of interest. This is evident from Fig. S1 (d) which demonstrates the improvement in AIS coverage between 2016 and 2019.

**Fig. S1 Temporal and spatial coverage of AIS data in rivers and 200 Nm coastal zone of China from 2016 to 2019.** (a) Daily evolution of AIS signals. (b) Spatial distribution of AIS signals in 2016. (c) Spatial distribution of AIS signals in 2019. (d) Spatial difference between 2016 and 2019.

**Ship technical specifications database (STSD)**

In this study, the extended Ship Technical Specification Database (STSD) was applied for ship emission calculation (Liu et al., 2016). The data from Lloyd's Register, China Classification Society (CCS) and Global Fishing Watch (GFW) (Kroodsma et al., 2018) were the most significant sources. In the current STSD, there are 101,638 ocean-going vessels (OGVs, defined here as vessel having an IMO number), a bit more than that recorded by United Nations Conference on Trade and Development (UNCTAD), e.g., 97.136 in 2019 (https://unctadstat.unctad.org/wds), which might due to the difference in OGV definition. The STSD provides static data which describes ship properties including vessel type, rated engine speed, rated engine power, length, width, height, design max speed, dead weight tonnage (dwt), maximum draught, build year, etc. Since STSD has incorporated data from GFW, CCS as well as Classification Societies of other East Asian countries, it also includes ships that are smaller than 500 Gt and usually don't have IMO numbers along China's coast, which take a large part in terms of the number of ships. However, the data is sometimes incomplete. Either excluding those particular ships from our computation or assigning default values to the property will lead to substantial inaccuracies. To correct the static data and reduce the error, we applied a machine learning method, Gradient Boosting Regression Tree (GBRT) to predict missing values based on other completed properties (Liu et al., 2016). This method previously applied for approximately 30% of the total ocean-going vessels in East Asia. However, as we updated the STSD to involved more than 350 thousand vessels, this kind of vessels only account for approximately 5% in terms of amount.

**Emission factors (EFs) in line with the DECA policy**

In this study, the average fuel sulfur content for heavy fuel oil (HFO) was assumed to be 2.43% and that for maritime gas oil (MGO) was 1.3%. These assumptions were consistent with our previous studies based on the investigation of global fleet from IMO Greenhouse Gas Study (IMO, 2014; IMO, 2020). The implementation of China's domestic emission control area (DECA) required low sulfur fuel, i.e., MGO with sulfur content <0.5% m/m for ships entering the area. Despite the mandatory date, some regions actually implemented the DECA policy ahead of time, such as Shenzhen port and ports in Yangtze River Delta (YRD). Meanwhile, fuel consumed by river vessels are demanded to use general diesel oil (GDO) with phased requirements, with sulfur content followed by 0.035%, 0.005% and 0.001%. Table S1 summarizes the actual performance of DECA from 2016 to 2019 and the corresponding fuel type in different area, including both the coastal seas and inland rivers. Emission factors for different fuel types are shown in Table S2, which were either directly obtained from Third IMO Greenhouse Gas study 2014 and related studies, or converted by the ratio of fuel sulfur content to the baseline, as illustrated in our previous work (Liu et al., 2016). In SEIM v2.0, a two-step method for was applied for ship emission estimation to be in line with the policy requirements, including the baseline EF selection and fuel correction factor (FCF) application:

- At the first stage, the baseline ship emissions were calculated for each two consecutive AIS signals based on the vessel's instantaneous power and the power-based baseline emission factors. The baseline EFs were selected according to fuel type of vessel recorded in STSD, including the liquified natural gas (LNG), HFO and MGO-0.13%. The GDO was only applied for river vessels, and the sulfur content for GDO was determined by time of AIS signals.
- In the policy-abutted modification module, the final ship emission would be further adjusted by the FCF. Due to the complexity of the DECA boundary, it would be time-consuming to judge whether it is in the DECA polygon for each AIS signal point. Thus, the intermediate output resulted from the first stage was grouped and aggregated by desired spatial resolution (e.g., 0.05° × 0.05°)

and other fields to reduce computing costs. For each aggregated emission record, vessels would be judged weather it was operating inside the DECA and needed to switch oil based on the signal time, geographical locations (latitude and longitude coordinates) and operating status. If the result of judgment is that the oil needs to be switched, the FCF, resulted from the quotient of the emission factors of the switched fuel and original fuel, would be further multiplied in the emission calculation formula.

| Fuel type         | Emission | Engine            | PM     | NO x        | NO x | NO x | SO 2 | HC  |
|-------------------|----------|-------------------|--------|------------------------|-----------------|-----------------|-----------------|-----|
|                   | Source   | type              |        | (Tier 0 d ) | (Tier I)        | (Tier II)       |                 |     |
| HFO
(2.43% S)  | ME       | $SSD^{a}$         | 1.335  | 18.1                   | 17              | 15.3            | 9.261           | 0.6 |
|                   |          | $MSD^{b}$         | 1.33   | 14                     | 13              | 11.2            | 10.215          | 0.5 |
|                   | AE       |                   | 1.339  | 14.7                   | 13              | 11.2            | 10.782          | 0.4 |
|                   | Boiler   |                   | 2.1    | 14.85                  | 14.85           | 14.85           | 0.1             | 0.1 |
| MGO
(0.05% S)  | ME       | SSD               | 0.31   | 17.01                  | 15.98           | 14.38           | 1.81            | 0.6 |
|                   |          | MSD               | 0.31   | 13.16                  | 12.22           | 10.53           | 1.98            | 0.5 |
|                   | AE       |                   | 0.32   | 13.82                  | 12.22           | 10.53           | 2.12            | 0.4 |
|                   | Boiler   |                   | 0.2    | 1.974                  | 1.974           | 1.974           | 3.1             | 0.1 |
| MGO
(0.13% S)  | ME       | SSD               | 0.199  | 17.01                  | 15.98           | 14.38           | 0.515           | 0.6 |
|                   |          | MSD               | 0.2    | 13.16                  | 12.22           | 10.53           | 0.568           | 0.5 |
|                   | AE       |                   | 0.202  | 13.82                  | 12.22           | 10.53           | 0.599           | 0.4 |
|                   | Boiler   |                   | 0.112  | 1.974                  | 1.974           | 1.974           | 0.825           | 0.1 |
| GDO
(0.035 %S) | ME       | SSD               | 0.0192 | 17.01                  | 15.98           | 14.38           | 0.133           | 0.6 |
|                   |          | MSD               | 0.0192 | 13.16                  | 12.22           | 10.53           | 0.147           | 0.5 |
|                   | AE       |                   | 0.0193 | 13.82                  | 12.22           | 10.53           | 0.155           | 0.4 |
| GDO
(0.05% S)  | ME       | SSD               | 0.0028 | 17.01                  | 15.98           | 14.38           | 0.019           | 0.6 |
|                   |          | MSD               | 0.0027 | 13.16                  | 12.22           | 10.53           | 0.021           | 0.5 |
|                   | AE       |                   | 0.0028 | 13.82                  | 12.22           | 10.53           | 0.022           | 0.4 |
| GDO
(0.001% S) | ME       | SSD               | 0.001  | 17.01                  | 15.98           | 14.38           | 0.004           | 0.6 |
|                   |          | MSD               | 0.001  | 13.16                  | 12.22           | 10.53           | 0.004           | 0.5 |
|                   | AE       |                   | 0.001  | 13.82                  | 12.22           | 10.53           | 0.004           | 0.4 |
| LNG               |          | Otto c | 0.03   | 1.3                    | 1.3             | 1.3             | 0.003           | 0.5 |

Table S2. Emission factors for different fuel types used in this study (Unit: g/kW·h)

a, b, c mean slow speed diesel engine (SSD), medium speed diesel engine (MSD) and Otto-cycle LNG-fueled engine, respectively.

dTier 0 refers to all ships constructed prior to January 1, 2000 which did not have an IMO Tier requirement at the time of construction.

**References**

*Fu, M., Liu, H., Jin, X., and He, K.: National- to port-level inventories of shipping emissions in China, Environmental Research Letters, 12, 114024, 10.1088/1748-9326/aa897a, 2017.*

Liu, H., Fu, M., Jin, X., Shang, Y., Shindell, D., Faluvegi, G., Shindell, C., and He, K.: Health and climate impacts of ocean-going vessels in East Asia, Nature Climate Change, 2016.

- Liu, H., Jin, X., Wu, L., Wang, X., Fu, M., Lv, Z., Morawska, L., Huang, F., and He, K.: The impact of marine shipping and its DECA control on air quality in the Pearl River Delta, China, Science of the Total Environment, 625, 1476-1485, 2018.
- D. Lack, B. Lerner, C. Granier, T. Baynard, E. Lovejoy, P. Massoli, A. R. Ravishankara and E. Williams: Light absorbing carbon emissions from commercial shipping, Geophysical Research Letters, 35, 10.1029/2008g1033906, 2008
- Liu, H., Jin, X., Wu, L., Wang, X., Fu, M., Lv, Z., Morawska, L., Huang, F., and He, K.: The impact of marine shipping and its DECA control on air quality in the Pearl River Delta, China, Science of the Total Environment, 625, 1476-1485, 2018.
- X. Zhang, Y. Zhang, Y. Liu, J. Zhao, Y. Zhou, X. Wang, X. Yang, Z. Zou, C. Zhang, Q. Fu, J. Xu, W. Gao, N. Li and J. Chen: Changes in the SO2 Level and PM2.5 Components in Shanghai Driven by Implementing the Ship Emission Control Policy, Environ Sci Technol,53, 11580-11587, 10.1021/acs.est.9b03315, 2019
- Z. Zou, J. Zhao, C. Zhang, Y. Zhang, X. Yang, J. Chen, J. Xu, R. Xue and B. Zhou: Effects of cleaner ship fuels on air quality and implications for future policy: A case study of Chongming Ecological Island in China, Journal of Cleaner Production, 267, 10.1016/j.jclepro.2020.122088, 2020
- J. Zhao, Y. Zhang, A. P. Patton, W. Ma, H. Kan, L. Wu, F. Fung, S. Wang, D. Ding and K. Walker: Projection of ship emissions and their impact on air quality in 2030 in Yangtze River delta, China, Environ Pollut, 263, 114643, 10.1016/j.envpol.2020.114643, 2020

**Q2. Ship emission impact**

Could the authors comment on how much the emission from ships actually affects air pollution in populated areas in China? They quote percentage contribution to emissions but a more useful number would be how concentrations are affected. I realise this is not part of this studies but maybe there is some literature on the subject?

**Response:**

Thank you for your comments. In this study we give a high-resolution ship emission inventory from 2016-2019, but air quality simulation is not the topic of this study. However, there has been a significant number of studies on impacts of shipping emissions on air pollution in China during previous decades. Some of results are summarized as follows:

- On a national scale, ship emissions increased the annual average  $PM_{2.5}$  concentration in eastern China to 5.2 µg/m-3, while the influence range of coastal ships can reach as far as 960 km inland (*Lv et al.*, 2018).
- At the regional scale, the average annual  $PM_{2.5}$  concentration in the Bohai Rim Area (BRA) increases by 5.9% due to ship emissions, and the contribution in summer is as high as 12.5% (*Chen et al., 2018*); shipping emissions contributed 0.60 µg/m-3 to the land ambient  $PM_{2.5}$  in Pearl River Delta (PRD) region, and the maximum contribution reached up to 2.54 µg/m-3 in Hong Kong (*Liu et al., 2018*); the annual contribution of ship emissions to  $PM_{2.5}$ concentration in the Yangtze River Delta (YRD) could reach 4.62 µg/m-3 in summer (*Feng et al. 2019*).
- At the port scale, previous studies have showed that ships contributed 20-30%

to  $PM_{2.5}$  in the coastal and riverside areas of Shanghai during the influence of ship plume (*Fan et al. 2016*); the contribution rate of ship emissions to  $PM_{2.5}$  concentration in summer in Qingdao is as high as 13.1%, while the that near the port may exceed 20% (*Chen et al. 2017*).

In the revised manuscript, this part has been reorganized to underline the effect of ship emission on air pollution in populated areas in China.

**Revisions in Manuscript:**

Line 48-55: The influence of coastal ships to annual average  $PM_{2.5}$  concentration (> 0.1 µg/m3) can reach as far as 960 kilometers inland in China (Lv et al., 2018). Exhaust emissions from ships contributed significantly to air pollution in major port clusters, e.g., Bohai Rim Area (BRA), Yangtze River Delta (YRD) and Pearl River Delta (PRD) regions, and the maximum increase of annual  $PM_{2.5}$  concentrations reaches up to 2 ~ 5 µg/m3, with the greatest impact on YRD region (Chen et al., 2018; Liu et al., 2018; Lv et al., 2018; Feng et al., 2019). During ship-plume-influenced periods, ships can even contribute over 20% of the total  $PM_{2.5}$  concentrations in port centers, e.g., Shanghai Port, Qingdao Port (Fan et al., 2016; Chen et al., 2017b).

**References:**

- Lv, Z., Liu, H., Ying, Q., Fu, M., Meng, Z., Wang, Y., Wei, W., Gong, H., and He, K.: Impacts of shipping emissions on PM2.5 pollution in China, Atmos. Chem. Phys., 18, 15811-15824, 10.5194/acp-18-15811-2018, 2018.
- Chen, D., Zhao, N., Lang, J., Zhou, Y., Wang, X., Li, Y., Zhao, Y., and Guo, X.: Contribution of ship emissions to the concentration of PM2.5: A comprehensive study using AIS data and WRF/Chem model in Bohai Rim Region, China, The Science of the total environment, 610-611, 1476, 2018.
- Feng, J., Zhang, Y., Li, S., Mao, J., Patton, A. P., Zhou, Y., Ma, W., Liu, C., Kan, H., Huang, C., An, J., Li, L., Shen, Y., Fu, Q., Wang, X., Liu, J., Wang, S., Ding, D., Cheng, J., Ge, W., Zhu, H., and Walker, K.: The influence of spatiality on shipping emissions, air quality and potential human exposure in the Yangtze River Delta/Shanghai, China, Atmos. Chem. Phys., 19, 6167-6183, 10.5194/acp-19-6167-2019, 2019.
- Liu, H., Jin, X., Wu, L., Wang, X., Fu, M., Lv, Z., Morawska, L., Huang, F., and He, K.: The impact of marine shipping and its DECA control on air quality in the Pearl River Delta, China, Science of the Total Environment, 625, 1476-1485, 2018.
- Chen, D., Wang, X., Nelson, P., Li, Y., Zhao, N., Zhao, Y., Lang, J., Zhou, Y., and Guo, X.: Ship emission inventory and its impact on the PM 2.5 air pollution in Qingdao Port, North China, Atmospheric Environment, 166, 351-361, 10.1016/j.atmosenv.2017.07.021, 2017.
- Fan, Q., Zhang, Y., Ma, W., Ma, H., Feng, J., Yu, Q., Yang, X., Ng, S. K., Fu, Q., and Chen, L.: Spatial and Seasonal Dynamics of Ship Emissions over the Yangtze River Delta and East China Sea and Their Potential Environmental Influence, Environ Sci Technol, 50, 1322-1329, 10.1021/acs.est.5b03965, 2016.

**Q3. Resolution of inventory**

The authors consistently talk about the high spatial and temporal resolution inventory

without actually stating clearly what the resolutions are. Please add this prominently in the manuscript.

**Response:**

Thank you for your kind reminder. The spatial resolution is 0.05° longitude and latitude grid, as illustrated in Figure 11, Figure 12, Figure 13, and the temporal evolution could be traced down to one single day (Figure 10) in this study. This information has been added to the Abstract and the Introduction section.

**Revisions in Manuscript:**

- 1) Lines 13-15: In this model, NOx, SO2, PM and HC emissions from ships in China's inland rivers and the 200 Nm coastal zone were estimated in every single day with a spatial resolution of  $0.05 \times 0.05$  degrees, based on a combination of Automatic Identification System (AIS) data and the Ship Technical Specifications Database (STSD).
- Lines 83-85: In this study, we developed the ship emission inventory (0.05° × 0.05°, daily) for the inland rivers and the 200 Nm coastal zone of China from 2016 to 2019 based on the global AIS data and the updated version of Shipping Emission Inventory Model (SEIM v2.0).

**Minor editorial points:**

**Q4. Line 8: pollutions should be pollution.**

**Response:**

Thank you for the comment. The word has been corrected.

**Revisions in Manuscript:**

Line 8: Ship emissions and coastal air **pollution** around China are expected to be alleviated with the gradually implemented of domestic ship emission control (DECA) policy.

**Q5. Line 107: use different language to 'figure out'.**

**Response:**

Thank you for the comment. We adopted the word "fill" instead of "figure out".

**Revisions in Manuscript:**

Line 117: The 10-minute linear interval interpolation method was used to figure out-fill long-distance AIS signal gaps.

**Q6. Line 129: 'hardly unified standard' does not make sense.**

**Response:**

Thank you for the comment. We have modified this expression.

**Revisions in Manuscript:**

Lines 142-143: Due to the complexity brought by the inconsistency of the ships' flag state, operating country and activity location, there is hardly unified standard it is hard to determine the attribution country of ship emissions.

**Q7. Line 150: replace 'averagely' with 'on average'.**

**Response:**

Thank you. The word has been corrected.

**Revisions in Manuscript:**

Line 164: The global dynamic AIS data for the whole year of 2016-2019 (from January 1st to December 31st) with averagely on average 30 billion signals per year, include both satellite-based signals and terrestrial-based signals, were collected to build a ship activity database.

**Q8.** Line **381**: 'proportions' does not need the 's'.**

**Response:**

Thank you. The word has been corrected.

**Revisions in Manuscript:**

Line 440: The **proportions** of ship  $SO_2$  emission from 12-50 Nm rose from 17.5% in 2016 to 35.3% in 2019, becoming the major spatial contributor in 2019.

**Q9.** Line 472: replace 'on' with 'to' and 'effect' with 'effects'.**

**Response:**

Thank you. These words have been corrected.

**Revisions in Manuscript:**

Line 535: This reminds us to pay attention on to additional environmental effect effects brought by detouring ships during the continuous implementation of DECA 2.0 policy.

**Q10. Throughout the whole manuscript NOx needs to have a subscript x.**

**Response:**

Thank you. The "NOx" has been revised throughout the whole manuscript.

---

## Author Comment (AC2)

**Response to Reviewers #2's Comments**

**Review of the paper**

Annual ship emissions around China under gradually promoted control policies from 2016 to 2019
by Xiaotong Wang and co-authors

The paper describes changes in shipping emissions around the Chinese coast between 2016 and 2019. The existing and previously published ship emission inventory model (SEIM) was updated and applied to AIS ship position data for 4 years and the results are analyzed for the effects of policy measures on ship emissions for different ship types and different distances to the coast.

The paper presents interesting results that can be of use for subsequent air quality simulations but also for policy measures to further reduce ship emissions along the Chinese coast. The papers needs significant language improvements, it is sometimes quite difficult to understand what the authors want to say. This should be done by a native speaker or through a professional language check. I will not mention all sentences that need improvements and clarification, because they are simply too many. However, these corrections need to be done before the paper can be published.

The paper also suffers from imprecise descriptions and some open questions concerning the results, but those can be treated in a revision. I recommend publication of this article in ACP after moderate revisions of the contents and major revisions of the language.

**Response:**

Thank you very much for the positive comment and interest in our work. All the comments are very professional and valuable for the revision and improvement of our research. We have carefully addressed all the comments and revised the manuscript accordingly. All grammatical problems in the revised manuscript have been carefully checked and vague expressions have been clarified to the most extent. As a matter of fact, we have learned that the language of this article will be polished and checked again before publication on Atmospheric Chemistry and Physics. The correction details for other problems are listed below point by point.

**Major comments:**

**Q1. Lines 8-28:**

In the abstract and throughout the paper it needs to said clearly which area is considered, when relative emission changes are given. I assume that the numbers mostly refer to the 200 nm zone along the Chinese coast, however, the area under investigation is much bigger, as shown in Fig. 11.

**Response:**

Thank you for your reminder. This study aims to evaluate the ship emissions targeting at the inland rivers and 200 Nm zone along the coast of China. Except for special explanation, the statistical results are all carried out for the inland rivers and the 200 Nm zone. As we applied a double-nested domain for emission estimation (see Q3 for detailed explanation), the spatial maps of shipping emissions (Fig. 5, Fig. 10, Fig. 11 and Fig. 12) could be given according to the domain 2 in Fig. 2.

**Revisions in Manuscript:**

1) Lines 13-15: In this model, $NO_x$, $SO_2$, PM and HC emissions from ships in **China's inland rivers and the 200 Nm coastal zone** were estimated in every single day with a spatial resolution of $0.05 \times 0.05$ degrees, based on a combination of Automatic Identification System (AIS) data and the Ship Technical Specifications Database (STSD).

2) Lines 143-145: In this study, the target area for developing ship emission inventory is **the navigable inland rivers and the coastal waters approximately within 200 Nm away from the Chinese mainland's territorial sea baseline (hereinafter referred to as 200 Nm zone),** as shown in Fig. 2.

**Q2. Lines 76-89:**

The number of 30 billion AIS signals does not say much about how complete the data is. You should say something on possibly missing data during certain times and how homogeneous the data is in time and space. In addition, you should give some information on the STSD, even though it might be described elsewhere, already. This could be done in the supplementary material. It is essential to know which technical information about the ships is typically available and which not. It is also surprising that 3.5 million vessel profiles are included given the fact that the number of large ocean going vessels is typically given as approx. 100,000. Is the rest of the data sets about small fishing boats (which won't have AIS in most cases) or about Chinese river vessels? This information can also be provided in section 2.3.

**Response:**

Thank you for your suggestions. These clear instructions helped improve our research and is very valuable for the revision.

**(1) AIS data**

The 30 billion AIS signals is the amount for global AIS data, which include both satellite-based signals and terrestrial-based signals. As we reduced the data and removed redundant signals, the data used in this study for China is about 100

million. We examined the AIS coverage and how homogeneous the data is in time and space after reducing. Results show that there exist missing signals or anomalies in particular days (Fig. S1), but the coverage of AIS data coverage has improved from 2016 to 2019. After the adoption of the 10-minute interpolation method, the AIS signal is expanded to about twice the original, and some periods with long intervals have been obviously supplemented. This part was added in the Supplementary Methods (see specific revisions below).

**(2) STSD Ship technical specifications database (STSD)**

At the same time, we carefully checked the total number of ships STSD covered and confirmed that the number of ship profiles included is about 350 thousand instead of 3.5 million. We assume that there should be a clerical error in the previous manuscript. The major sources of STSD are data from Lloyd's Register, China Classification Society (CCS) and Global Fishing Watch (GFW). And technical specifications of ships include MMSI number, IMO number, length, width, draft, rated power, built year, type, max speed, registered country, engine speed, dead water tonnage, etc. One of the improvements of STSD in this study is that it has been enriched, incorporated data from GFW, CCS as well as Classification Societies of other East Asian countries. We have cleaned the data by eliminating duplicated ships, ships with apparently faulty or too many missing properties. 350 thousand ships remained at last. In current STSD, there are 101,638 ocean-going vessels (OGVs, defined here as vessel having an IMO number), a bit more than that recorded by United Nations Conference on Trade and Development (UNCTAD), e.g., 97,136 in 2019 (https://unctadstat.unctad.org/wds), probably due to the difference in OGV definition. Other ships are mostly smaller than 500 Gt and usually don't have IMO numbers. In the revised manuscript, we also added a brief introduction to STSD in the Supplement.

**Revisions in Manuscript:**

1) Lines 85-87: The global AIS database with annually ~30 billion signals, together with Ship Technical Specifications Database (STSD) covering over **350 thousand individual vessels** were combined as fundamental data for emission calculation.

2) Lines 164-166: The global dynamic AIS data for the whole year of 2016-2019 (from January 1st to December 31st) with on average 30 billion signals per year, include both **satellite-based signals and terrestrial-based signals,** were collected to build a ship activity database.

3) Lines 175-180: After reducing, the AIS coverage in our study area has been examined in terms of time and space **(see Supplementary Methods and Fig. S1).** Short period drops probably result from missing or abnormal AIS signals

due to many reasons, such as disruption to satellites, equipment maintaining, data transmission fault, ships sailing beyond terrestrial station receiving range etc, which is a common phenomenon that has been pointed out by previous studies (Goldworthy et al., 2019; Johansson et al., 2017; IMO, 2020). To assure the reliability of total emissions, it's important to have whole year data instead of using several weeks and then multiplied to annual total.

4) Lines 182-187: The STSD describes ship properties such as vessel type, dead weight tonnage (DWT) and engine power, designed speed, flag state, etc., which has also been updated to 2019. The extended STSD currently contains over 350 thousand vessels, in which 101,638 are OGVs, which is consistent with the statistics of the United Nations (UNCTAD, 2019). Besides the ship data collected from Lloyd's Register and the Classification Societies of various countries, we have also incorporated fishing ships and smaller ships that don't have IMO numbers from Global Fishing Watch (GFW) (Kroodsma et al., 2018). These ships were observed to be quite active along China's coast. A further introduction to the updated STSD was provided in the Supplementary Methods.

**Revisions in Supplement:**

**Automatic Identification System (AIS) data**

The temporal and spatial coverage of AIS data were examined to guarantee the quality of ship emission inventories. The full year AIS data including both satellite signals and territorial signals from 2016 to 2019 were used for our emission calculations in this study. Fig. S1a showed the homogeneity of the AIS signals in this study in terms of time. It is noticeable that during February (Spring Festival Holiday in China) and May to August (Fishing-off Season in China), the number of daily AIS signals is lower than average (approx. 5 million/day). Missing signals or anomalies occasionally exist, which could due to multiple factors, such as disruption to satellites, equipment maintaining, data transmission fault etc., Besides, Bad weather could be a reason for interference of signal transmission. After the adoption of the 10-minute interpolation method, the AIS signal is expanded to about twice the original, and some periods with long intervals have been obviously supplemented. Fig. S1 (b) and (c) exhibited the change of spatial coverage of AIS signals in inland waters and coastal waters around China. The number of AIS messages transmitted per year is increasing over the span of this study's years of interest. This is evident from Fig. S1 (d) which demonstrates the improvement in AIS coverage between 2016 and 2019.

**Ship technical specifications database (STSD)**

In this study, the extended Ship Technical Specification Database (STSD) was applied for ship emission calculation (Liu et al., 2016). The data from Lloyd's Register, China Classification Society (CCS) and Global Fishing Watch (GFW) (Kroodsma et al., 2018) were the most significant sources. In the current STSD,

there are 101,638 ocean-going vessels (OGVs, defined here as vessel having an IMO number), a bit more than that recorded by United Nations Conference on Trade and Development (UNCTAD), e.g., 97,136 in 2019 (https://unctadstat.unctad.org/wds), which might due to the difference in OGV definition. The STSD provides static data which describes ship properties including vessel type, rated engine speed, rated engine power, length, width, height, design max speed, dead weight tonnage (dwt), maximum draught, build year, etc. Since STSD has incorporated data from GFW, CCS as well as Classification Societies of other East Asian countries, it also includes ships that are smaller than 500 Gt and usually don't have IMO numbers along China's coast, which take a large part in terms of the number of ships.

[Figure]

**Fig. S1 Temporal and spatial coverage of AIS data in rivers and 200 Nm coastal zone of China from 2016 to 2019. (**a) Daily evolution of AIS signals. (b) Spatial distribution of AIS signals in 2016. (c) Spatial distribution of AIS signals in 2019. (d) Spatial difference between 2016 and 2019.

**Q3. Line 104/105:**

What is the purpose of the "double nested domain"? What are the potential "boundary effects"?

**Response:**

The "double-nested domain" method was introduced in the Supplementary Information of our work published in 2016 (Liu et al., 2016). To put it simply, "boundary effects" refer to the sharp increase/decrease on the boundary when calculating the emission inventory in a defined region. The "double nested domain"

is dedicated to solving the error caused by the "boundary effects".

The improvements can be explained by two example cases, as indicated by the green and purple curves in **Fig. Q3.1** below. For case 1, a ship moves following the green curve and sends AIS signals at point A and point B. If only D2 was used for calculation, the starting point for this ship would be point B, which is the first AIS signal after entering the D2. Thus, the emissions from the boundary to point B would be overlooked. By using the new method with two nested domains, the actual emission between point A and B was calculated and evenly distributed along the straight line between A and B. Then the emissions from the boundary to point B were included for the grid boxes inside the region. In contrast, case 2 indicated by the purple line, would overestimate emissions in a region without applying the two-nested domain. The AIS data shows that some vessels voyage across the research boundary, going out of the region and then sailing back into the region. With only D2 is set, all the navigation time between AIS signal C and D would be used to calculate emissions and then the emission would be distributed along a straight line between points C and D (as shown by the dotted line). However, the truth is that most of the emissions happen outside the regional boundary. Our new method successfully avoids these errors by introducing an extra domain with broader boundaries.

[Figure]

**Fig. Q3.1 Schematic diagram of the method of double-nested domain to solve the boundary effect in AIS-based shipping emission inventory.**

In this revised manuscript, we added the citation of our previous work and added a brief introduction of the "boundary effects".

**Revisions in Manuscript:**

Lines 112-114: To reduce the uncertainties of emission calculation, we have previously introduced several techniques in SEIM v1.0 (Liu et al., 2016): 1) a double-nested research domain was applied to reduce the boundary effects (i.e., **sharp increase/decrease on the boundary when calculating the emission inventory in a defined region**); …

**Q4. Line 105/106:**

What is the GBRT method? In how many cases do the default values have to be estimated because of missing ship properties?

**Response:**

GBRT (Gradient Boosting Regression Tree) is a machine learning method used for predicting missing values of ship properties, e.g., engine power, dead weight tonnage, maximum designed speed, etc., based on the available information in Ship technical specifications database (STSD). Details of the GBRT method has also been introduced in the Supplementary Information of our previous work (Liu et al., 2016). This method previously applied for approximately 30% of the total ocean-going vessels in East Asia. However, as we updated the STSD to involved more than 350 thousand vessels, this kind of vessels only account for approximately 5% in terms of amount.

In the revised manuscript, we made a little revision to the original sentence.

**Revisions in Manuscript:**

Lines 114-116: 2) the Gradient Boosting Regression Tree (GBRT) method was adopted to  **predict missing values of ship properties**;

**Revisions in Manuscript:**

**Ship technical specifications database (STSD)**

…However, the data is sometimes incomplete. Either excluding those particular ships from our computation or assigning default values to the property will lead to substantial inaccuracies. To correct the static data and reduce the error, we applied a machine learning method, Gradient Boosting Regression Tree (GBRT) to predict missing values based on other completed properties (Liu et al., 2016). This method previously applied for approximately 30% of the total ocean-going vessels in East Asia. However, as we updated the STSD to involved more than 350 thousand vessels, this kind of vessels only account for approximately 5% in terms of amount.

**Q5. Line 111-114:**

This is a nearly exact repetition of a text from the introduction. You should avoid such repetitions.

**Response:**

Thank you for your kind reminder. We have modified the similar sentences in the introduction section. We will tend to avoid repetitions in following works.

**Revisions in Manuscript:**

Lines 87-88: The technical details of upgrading the previous SEIM v1.0 to SEIM v2.0 were introduced in the Methods.

**Q6, Line 118:**

How long is "long time gap"? So, in which cases is the restoration method applied?

**Response:**

Thank you for the question. The "long time gap" we mentioned previously might not be accurate. The application situations and implement steps have been described by a sketch map in section 2.4.1 and Fig. 4. For each two consecutive AIS point A and B, if line AB intersects the continent and is not contained in the continent (considering this might be inland river), the route restoration method would be applied. However, the complex coastline makes it rather time-consuming to judge the geographical relationship between the trajectory line and the continent polygon one by one, we thus added an additional distance threshold of 50 km in the model, i.e., the restoration method would only be applied **when the distance of "cross-land trajectory" is over 50 km**. This setting would skip some cases when ships were sailing in the estuaries, crossing the coastlines.

In the revised manuscript, the "lone time gap" was replaced with "cross-land trajectory with long distance", and the statement of distance threshold was added to section 2.4.1.

**Revisions in Manuscript:**

1) Lines 131-133: Second, a route restoration module is applied for  **cross-land trajectory with long distance** in AIS data, in which the 10-minute linear interpolation will be applied on the shorted paths instead.

2) Lines 237-241: However, as it was rather time-consuming to judge the geographical relationship between the trajectory line and the continent polygon, an additional distance threshold of 50 km was finally added in the model, i.e., the restoration method would only be applied for "cross-land trajectory with long distance". This setting would skip some cases when ships

were sailing in the estuaries, crossing the coastlines.

**Q7. Line 160 and Table 1:**
How are the total main engine power and the total dead weight tonnage calculated?

**Response:**

Thank you for your question. The main engine power and dead weight tonnage (DWT) for each particular ship are recorded in the STSD. The total main engine power and total DWT in **Fig. 3** and **Table 1** are both the accumulated values of all ships appears in the 200 Nm zone of China for each year, regardless of the counts of AIS signals or voyages. To avoid misleading, we removed the statistical result regarding the main engine power and total dead weight tonnage in the revised manuscript. The number of vessels and total operating hours in Table 1 would be more useful to reflect the ship activities.

**Revisions in Manuscript:**

Lines 191-195: Table 1 shows the statistical results of the AIS messages and active ships for different years in this study. From 2016 to 2019, an annual average of about 90,000 vessels were observed in in inland rivers and the 200 Nm zone of China, and the number of vessels showed a downward trend year by year.

**Table 1: Statistics of AIS messages and active ships in China in 2016–2019.**

| Statistical items | | 2016 | 2017 | 2018 | 2019 |
|---|---|---|---|---|---|
| Global | Archived AIS messages ($10^9$) | 26 | 35 | 31 | 45 |
| | Active ships with unique MMSI ($10^3$) | 523 | 635 | 754 | 824 |
| China (River and 200 Nm zone) | Number of identified ships ($10^3$) | 96 | 92 | 88 | 85 |
| | Total operating hours ($10^6$ hours) | 196 | 197 | 195 | 202 |

**Q8. Line 165 – 169 and Figure 3:**
In Figure 3 it looks like there are short periods of one or few days during which the activity drops significantly (e.g. mid June, beginning of July, beginning of August). What is the reason for this? Is this bad weather, non-working days or something similar? Or are these periods with missing AIS data?

**Response:**

Thank you for your question. We checked the AIS data used in this study and discovered that in different years, AIS gaps appeared at different time. This indicates that these short period drops probably result from missing AIS signals. Because AIS data are obtained from both Satellite-based AIS and terrestrial AIS. Disruption to satellites, equipment maintaining, data transmission fault, ships sailing beyond terrestrial station receiving range etc. can cause missing or abnormal signals in AIS data. The data gap is a common phenomenon that occasionally occurs in AIS, which has also been pointed out by previous studies (*Goldworthy et al., 2019; Johansson et al., 2017; IMO, 2020*). So, it's important to provide day-by-day results and then estimates the annual total, instead of picking several typical days or weeks. In this revision, we kept the Fig. 3 to give temporal profile of vessel operating time and added more explanations regarding the coverage of AIS data in both the manuscript and the Supplementary Methods.

**Revisions in Manuscript:**

Lines 175-180: After reducing, the AIS coverage in our study area has been examined in terms of time and space (see Supplementary Methods and Fig. S1). Short period drops probably result from missing or abnormal AIS signals due to many reasons, such as disruption to satellites, equipment maintaining, data transmission fault, ships sailing beyond terrestrial station receiving range etc, which is a common phenomenon that has been pointed out by previous studies (Goldworthy et al., 2019; Johansson et al., 2017; IMO, 2020). To assure the reliability of total emissions, it's important to have whole year data instead of using several weeks and then multiplied to annual total.

**Revisions in Supplement:**

**Automatic Identification System (AIS) data** (See Q2)

**References:**

*Goldsworthy, B., Enshaei, H., and Jayasinghe, S.: Comparison of large-scale ship exhaust emissions across multiple resolutions: From annual to hourly data, Atmospheric Environment, 214, 10.1016/j.atmosenv.2019.116829, 2019.*

*Johansson, L., Jalkanen, J.-P., and Kukkonen, J.: Global assessment of shipping emissions in 2015 on a high spatial and temporal resolution, Atmospheric Environment, 167, 403-415, https://doi.org/10.1016/j.atmosenv.2017.08.042, 2017.*

*IMO: Fourth IMO GHG Study - Final Report, CE Delft, 2020.*

**Q9. Line 182:**

You should consider that a method very similar to the route restoration method described here was already introduced by Aulinger et al. in 2016 (Atmos. Chem. Phys., 16, 739–758).

**Response:**

Thank you for your comments. We have read this work and supplement it to our

manuscript as a reference in Line. In the works done by Aulinger et al. (2016) and Johansson et al. (2017), route regeneration algorithm was applied to solve the long-time gap in consecutive AIS signals. Several factors would affect whether this method works well or not: AIS data's quality, the accuracy and fineness of predefined sea routes, time span between two consecutive interpolation points, etc. In this study, technical details of route regeneration algorithm applied around China sea was introduced in detail, including specific routes that are predefined, time gaps between interpolated points and the algorithm flow for peer review, comparison and reference.

**Revisions in Manuscript:**

Lines 214-215: Similar methods but with featured details has been previously experimented by **Aulinger et al. (2016) on a regional scale and Johansson et al. (2017) on a global scale.**

**References:**

*Aulinger, A., Matthias, V., Zeretzke, M., Bieser, J., Quante, M., and Backes, A.: The impact of shipping emissions on air pollution in the greater North Sea region – Part 1: Current emissions and concentrations, Atmospheric Chemistry and Physics, 16, 739-758, 10.5194/acp-16-739-2016, 2016.*

*Johansson, L., Jalkanen, J.-P., and Kukkonen, J.: Global assessment of shipping emissions in 2015 on a high spatial and temporal resolution, Atmospheric Environment, 167, 403-415, https://doi.org/10.1016/j.atmosenv.2017.08.042, 2017.*

**Q10. Line 222:**

What are the criteria for a method to be "basically satisfactory"?

**Response:**

Thank you for the comment. Our original intention is that the result of spatial distribution results of AIS signals of OGVs, CVs and RVs is basically consistent with experience, i.e., they showed different navigation range, with OGVs mainly at seas, CVs near the coast and RVs in inland waters. Thus, we modified this improper expression.

**Revisions in Manuscript:**

Lines 258-260: The spatial distribution of AIS signals of OGVs, CVs, and RVs **were basically consistent with experience, with OGVs mainly at seas, CVs near the coast and RVs in inland waters.** .

**Q11. Line 253:**

Can you say something about how well the emission inventory might agree with real world emissions? Despite possible methodological problems, which might be difficult to avoid (e.g. because of missing technical information about the ships), can you say something about the percentage of ships that do not follow the DECA rules (i.e. the

non-compliance rate)?

**Response:**

Thank you for your comments.

Two sources of uncertainty in regional shipping emissions estimation were considered: 1) the completeness of ship observations and 2) the estimates of annual emissions from the observed fleet of ships. In our previous work (Liu et al., 2016), a Monte Carlo method was also used to evaluate the uncertainty for our bottom-up emission inventories, which was estimated to be among 3% to 6%. In this study, completeness of both AIS data and STSD has been improved and SEIM has also been upgraded. These efforts all contributed to the consistency of the model to the real world and to some extent alleviated uncertainties. But inevitably there are still several uncertainties in this model, including AIS data gap and anomaly (influenced by methodological conditions, equipment maintaining, etc.), accuracy and coverage of STSD information, accuracy of RVs, CVs and OGVs classification, route restoration algorithm, obedience of ships under DECA policy, etc. We have added sentences in revised manuscript addressing the uncertainties of our emission inventory in this article.

Guaranteed by the authority of Chinese government, DECA policy was endowed it with the power to restrict the fuel ship-owners uses. Several investigations, using whether field measurements or simulations, have showed that with DECA policy, not only ship fuels have been found to be cleaner, but also air pollution caused by shipping activities have been evaluated to be less in important ports alongside Chinese coast *(Zhang et al., 2019b; Zhang et al., 2018a; Zou et al., 2020).* However, there are not sufficient evidence indicating all vessels stick to DECA's regulation or the violation rate of DECA policy from year to year. This could undoubtably cause uncertainties in emission estimation. In the revised manuscript, an explanation was added to address this potential uncertainty.

**Revisions in Manuscript:**

1) Lines 122-126: These improvements contributed to the consistency of the model to the real world and to some extent alleviated the uncertainties in our model. But inevitably several uncertainties still exist in this model, including AIS data gap and anomaly (influenced by methodological conditions, equipment maintaining, etc.), accuracy and coverage of STSD information, accuracy of RVs, CVs and OGVs classification, route restoration algorithm, obedience of ships under DECA policy, etc.

2) Lines 291-297: It is worth noting that as far as we know, there has not been

sufficient evidence showing all vessels are sticking to DECAs or the violation rate each year. But there are studies indicating the effectiveness of DECAs in recent years (Liu et al., 2018; Zhang et al., 2019; Zhao et al., 2020; Zou et al., 2020). Not only have fuels been found to be cleaner (Zhang et al., 2019), but also air pollution caused by shipping activities has been less in important ports alongside Chinese coast (Zou et al., 2020). Guaranteed by the authority of Chinese government, we assume that the DECA policy should mostly be effective, but lack of evidence about the violation of DECAs added to uncertainties in this model.

**References:**

Liu, H., Fu, M., Jin, X., Shang, Y., Shindell, D., Faluvegi, G., Shindell, C., and He, K.: Health and climate impacts of ocean-going vessels in East Asia, Nature Climate Change, 2016.

Zhang, X., Zhang, Y., Liu, Y., Zhao, J., Zhou, Y., Wang, X., Yang, X., Zou, Z., Zhang, C., Fu, Q., Xu, J., Gao, W., Li, N., and Chen, J.: Changes in the SO2 Level and PM2.5 Components in Shanghai Driven by Implementing the Ship Emission Control Policy, Environmental Science & Technology, 53, 11580-11587, 10.1021/acs.est.9b03315, 2019b.

Zhang, Y., Deng, F., Man, H., Fu, M., and Liu, H.: Compliance and port air quality features of ship fuel switching regulation: by a field observation SEISO-Bohai, Atmospheric Chemistry and Physics, 1-28, 2018a.

Zou, Z., Zhao, J., Zhang, C., Zhang, Y., and Zhou, B.: Effects of cleaner ship fuels on air quality and implications for future policy: A case study of Chongming Ecological Island in China, Journal of Cleaner Production, 267, 122088, 2020.

**Q12. Line 283/284:**

Can you say why the SO2 emissions you calculate might be higher than those in Li et al. (2018)? I understood that RVs have very low sulfur emissions which would means that they won't contribute a lot to the total emissions, even when they sail in coastal waters. Or could they switch fuel when leaving the river?

**Response:**

Thank you for your comments.

We would like to explain the fuel settings for RVs sailing in coastal waters first. In the SEIM v2.0 model, RVs were assumed to use general diesel oils (GDOs, sulfur content gradually decreased from 350 ppm to 10 ppm) complying with Chinese standards, so RVs would not switch fuels even when they sail in coastal waters. By the Article 65 of Law of the People's Republic of China on the Prevention and Control of Atmospheric Pollution issued in 2015 and revised version 2018, *it is prohibited to sell residual oil and heavy oil for non-road mobile machinery, as well as inland and river-to-sea vessels*. Thus, the fuel types and emission factors for RVs would keep unchanged regardless of the vessel location, within the period of the certain fuel standard.

As for the comparison with Li et al. (2018)'s study, the different of emission results for RVs might come from multiple aspects. Table Q12.1 lists the difference of RVs regarding the activity data, identification method of RVs, emission estimation method, emission factors and typical results. Due to the great differences in data sources and methods between this study and Li et al. (2018), it might be hard to make rigorous comparisons. We currently consider two major reasons causing the differences:

- We identified RVs based on AIS signals distribution in this study. Given the fact that CVs and even OGVs sail at inland waters sometimes, there exists possibilities that some CVs and OGVs are mistakenly identified as RVs, adding to the number of RVs in this study. Thus, the identified numbers of RVs might be higher than that in Li et al., (2018).
- We applied fuels with sulfur content complied with the national standards, for which the emission factors of $SO_2$ and PM would be much lower than CVs or OGVs. Thus, the emission shares of $SO_2$ and PM appeared to be lower in this study that in Li et al., (2018), but it was opposite for $NO_x$ and other pollutants.

Table Q12.1 Comparison of ship emission estimation of RVs in China.

|  | **This study** | **Li et al. (2018)** |
|---|---|---|
| Targe year | 2016-2019 | 2013 |
| Activity data | Satellite- and terrestrial-based global AIS data (approx. 90,000 vessels) | National and local maritime department (MD), AIS sample (700 vessels) |
| Identification method of RVs | AIS trajectory distribution | Registration information in MD |
| Emission estimation methods | For each AIS signal, estimate emissions by vessel installed engine power, transient load factor and the time interval to the next signal | For each vessel arrived at port, estimate emissions by vessel installed engine power, average load factor and operating time in different modes |
| Emission factor | Power-based emission factor (g/kW·h) | Fuel-based (g/kg fuel) emission factor and fuel consumption rate (g fuel/kW·h) |
| Sulfur content | 0.035% in 2016 | 0.5% in 2013 |
| Emission shares of RVs | $SO_2$ (<1%), PM (1.1%), $NO_x$ (13.2%), HC (19.3%) | $SO_2$ (2%), PM (3%), $NO_x$ (6%), HC (12%) |

Given the above reasons, we modified the explanations for the difference in the manuscript.

**Revisions in Manuscript:**

Lines 329-337: The emission shares of RV may differ from that by Li et al. (2018), considering the two major reasons. One the one hand, as we identified RVs based on spatial frequency distribution of ship trajectories in AIS, which allows vessels sometimes operating in coastal waters. Given the fact that CVs and even OGVs sail at inland waters sometimes, there exists possibilities that some CVs and OGVs are mistakenly identified as RVs. Thus, the identified vessels of RVs might be higher than that in Li et al., (2018). One the other hand, since we applied GDOs with sulfur content up to the national standard to RVs, for which and the emission factors of $SO_2$ would be much lower, the emission shares of $SO_2$ appeared to be lower than that in Li et al., (2018), but it was opposite for $NO_x$ and other pollutants.

**References**

Li, C., Borken-Kleefeld, J., Zheng, J., Yuan, Z., Ou, J., Li, Y., Wang, Y., and Xu, Y.: Decadal evolution of ship emissions in China from 2004 to 2013 by using an integrated AIS-based approach and projection to 2040, Atmos. Chem. Phys., 18, 6075-6093, 10.5194/acp-18-6075-2018, 2018.

**Q13. Line 295-308.**

In my opinion, this investigation w.r.t. flag state does not add much information. Because shipping is international and the flag state does not even say something about the vessel owner, this analysis does not tell much.

**Response:**

Thank you for your comments. We adopted your suggestion and removed the analysis of shipping emission contributed by different flag states in the original section 3.1.2, as well as corresponding results in Abstract and Discussion. At the same time, we retained the comparison of OGV emission in the 200 Nm zone of China v.s. global shipping emissions and inserted into the original section 3.1.1 (now 3.1).

**Revisions in Manuscript:**

1) Lines 324-327: Compared to a recent estimation of global ship emissions (IMO, 2020), it is striking that OGVs in the 200 Nm zone of China contributed to 9.7 ~ 14.3% of global OGV emission (Table S3), despite only <1% of the world's sea area. Such result suggests the substantial concentration of shipping emissions from global fleet around China.

2) Line 22: .

3) Line 520: .

**Q14. Figure 9:**

Short term drops in emissions are apparent in the time series, which do not correspond to the spring festival, e.g., mid of 2017, approx. Sept 2018. What is the reason for them?

**Response:**

Thank you for the question. The short-term drops in emissions for certain days (Jan. 1, 2017 and Aug. 1-9, 2018) are possibly caused by the overlong time gap in AIS data. To illustrate this, we put the time series of AIS signals and the emissions together, as shown in **Fig. Q14.1**. It can be observed that there are occasionally missing AIS signals in the whole time series, which might due to occasional disruption to satellites, equipment maintaining, data transmission fault etc., as we explained in **Q8**. We consider this the systematic error of the method, but it generally has minor influence on daily variations of ship emissions from 2016 to 2019. Moreover, the gap of AIS signal will not lead to the same degree of reduction of ship emissions, as the 10-minute interpolation method in SEIM v2.0 model could effectively avoid the emission deviation caused by duplicate or missing AIS signals.

[Figure]

**Fig. Q14.1 Daily changes of AIS signals and ship emissions of China in 2016-2019. The** *y*=1 **line refers to the daily average value of 2016 for each item.**

In the revised version, we added a supplementary figure to show the temporal coverage of AIS data and added corresponding descriptions (see **Q2** for revisions).

**Q15. Figure 10:**

What is the reason for the steep short-term increase in emissions in some ports in 2018 (Tianjin, Ningbo Zhoushan, Shenzhen)?

**Response:**

Thank you for your question. We checked the data and found that the steep short-term increase of ship emissions in Tianjin, Ningbo-Zhoushan and Shenzhen ports occurred during approximately Sep. 2-25, 2018. This may be due to the change of ship dynamic information (e.g., navigation speed) caused by the interference of external factors (e.g., typhoon), resulting in upward biases of ship emission

estimation.

First, we checked the number of AIS signals and operating time of ships for these ports, but no sharp increase was found during the same period, when steep short-term increase was noticeable in $SO_2$ emissions. This eliminated the possibility of sudden change in ship activities being the reason for unusual $SO_2$ emissions. Second, we checked the time series of ship emissions and operating time from multiple aspects, e.g., vessel type, vessel age, flag country, size bin, operating mode, and found that the steep short-term increase was mostly related to the vessel type and operating mode. As shown in **Fig. Q15.1(a)**, the emissions from containers increased significantly during Sep. 2-25, 2018, while that from bulk carriers and tankers slightly increased. When divided according to the operating status, we found that emissions from containers at sea and maneuvering were higher than before, while that from ships at berth were lower, as illustrated in **Fig. Q15.1(b)**. The operating time of containers under different operating modes present similar characteristics, as shown in **Fig. 15.1(c).** Due to the relatively high main engine power, container ships would generate higher emission intensity under cruising mode compared to berthing, during the same operating time. Thus, the ship emissions for containers appeared to increase more significantly during Sep. 2-25, 2018, compared to other vessel types **(Fig. 15.1a)**. Moreover, major container ports such as Tianjin port, Ningbo-Zhoushan port and Shenzhen port were also observed to have more notably emissions during Sep. 2-25, 2018, among all ports in **Fig. 9** in the manuscript.

Given the fact that the operating mode was defined by navigation speed and load condition, we speculate that the most likely reasons causing the unusual change in operating mode would be the abnormal dynamic data (e.g., speed) in AIS signals, which could somehow be interfaced by external factors. According to China's meteorological data (http://www.typhoon.org.cn/), "Super Typhoon Mangkhut" formed on the Northwest Pacific Ocean in early September in 2018 and landed in China in mid-September. We speculate that the navigation speed of vessels might be affected by the typhoon passing through, thus affecting the judgement of operating mode in the model, i.e., ships at berth or anchorage were recognized as maneuvering or cruising. Due to the great difference between the emission intensities of ships in berthing and cruise mode (especially container ships), the emissions increased sharply from Sep. 2 to 25. However, more evidence is needed to verify the influence of extreme meteorological conditions on AIS signal.

[Figure]

**Fig. Q15.1 The 5-day moving average of SO₂ emissions for Tianjin, Ningbo-Zhoushan and Shenzhen port in 2018. (a) Ship emissions from typical vessel types. (b) Container emissions classified by operating mode. (c) Operating time of containers under different modes.**

The calculation method has been applied in the whole period from year 2016-2019, but the steep short-term increase in emissions only appeared in certain period and certain ports. However, smoothing or removing could not guarantee that these biases did not occur in other time periods. They could neither be simply corrected as the true value is not known. So, we kept the original results without correction while admitting the uncertainties of the method on the fine time scale.

**Revisions in Manuscript:**

Lines 414-418: In addition, steep short-term increases in SO₂ emissions were observed for Tianjin, Ningbo Zhoushan, Shenzhen ports in September, 2019. These peaks were speculated to be due to the inaccurate vessel dynamic information in AIS signals caused by the interference of adverse weather, i.e., "Super Typhoon Mangkhut". However, more evidence is needed to verify the influence of extreme meteorological conditions on AIS signal.

**Q16. Line 336/337:**

Could you explain which role wind direction plays for ship activities?

**Response:**

Thank you for your question. Ship activities are mostly scheduled, which could somehow be affected by extreme events or weather, but they are not supposed to be strongly affected by the wind direction. In terms of the ship emissions, wind direction could act as a resistance or a propulsion to dynamically affect the ship emission intensity. Under headwind conditions, additional power was wasted to counter the wind power, so the fuel consumption of shipping increases and emissions rise. But this impact is independent of the ship's activities. Taking these factors into account, in the revised manuscript we removed this misquoted sentence from Chen et al. (2019b). Thank you again for your careful review valuable comments.

**Revisions in Manuscript:**

Line 412: …ship emissions in PRD region were higher in in spring and summer since wind direction were more advantageous for ship activity in spring and summer (Chen et al., 2019b);…

**Q17. Line 374:**

How can the improved emissions reduce uncertainties in an air quality model? It might improve the results of an air quality model application.

**Response:**

Thank you for pointing out this vague expression. We revised the sentence according to your suggestion.

**Revisions in Manuscript:**

Lines 432-433: This spatial improvement of ship emissions with the route restoration method is expected to reduce uncertainties in the air quality model **improve the results of an air quality model application**.

**Q18. Line 398:**

I cannot see how a North Atlantic shipping route would be visible in this emission inventory. Perhaps you mean a North Pacific route? A similar argument holds for the mentioned Asia-Europe routes: Do you know from the AIS data where the ships are heading?

**Response:**

Thank you for your kind reminder. The "North Atlantic Route" is a mistake and what we meant is indeed the "North Pacific Route". To illustrate the major routes of OGVs, we provide Fig. Q18.1 to show the spatial distribution of AIS signals in Domain 1 of the study area. Although the satellite-based signals in the open sea may be sparse than terrestrial -based ones closer to the coast, it is still clearly visible

that higher density of AIS signals on major routes could extend eastward to the Pacific Ocean, which is regarded the North Pacific Route. However, we also found that Asia-Europe routes stated here may not be proper, as it only show the legs between Chinese ports to Malacca Strait. Thus, in the revised manuscript, we changed "North Atlantic Route" to "North Pacific Route", and replaced the "Asia-Europe routes" with "routes from Chinese ports to Malacca Strait".

[Figure]

**Fig. Q18.1 Spatial distribution of AIS signals in Domain 1 of the study area.**

**Revisions in Manuscript:**

Lines 458-460: The most intensive near-sea routes included China-Korea, China Mainland-Taiwan, the North  **Pacific** Route,  **routes from Chinese ports to Malacca Strait** and routes between busy ports of China, such as main ports in BRA, YRD and PRD (Fig. 12a).

**Q19. Line 424:**

There is just one scenario defined (the No-DECA scenario). I would not call the emission calculation for the years 2016-2019 that consider all regulations in place a scenario. Therefore, you do not define "another" scenario.

**Response:**

Thank you for your kind reminder. We replace "another scenario" to "a scenario" throughout the manuscript.

**Revisions in Manuscript:**

1) Line 93: In addition,  **a** scenario without the DECA policy was performed to evaluate the effect of China's gradually implemented DECA

policy, considering the actual change of interannual ship activities.

2) Line 300: …we designed  **a** scenario (No-DECA scenario) in SEIM v2.0, as listed in Table 2.

3) Line 485: …we designed  **a** scenario without DECA policy to evaluate the emission reduction effect considering the annual change of ship activities.

**Q20. Line 455:**

Here you should make clear that the numbers refer to the 200 nm zone (if I am not mistaken):

**Response:**

Thank you for your kind reminder. We have added "in 200 Nm zone of China" in this sentence.

**Revisions in Manuscript:**

Lines 516-517: As a result, $SO_2$ and PM emissions from ships decreased by 29.6% and 26.4%, respectively, **in 200 Nm zone of China** in 2019 compared to 2016.

**Q21. Line 471/472:**

The sentence is repeated from lines 393/394. You should avoid such replications

**Response:**

Thank you for your kind reminder. We rewrote this sentence.

**Revisions in Manuscript:**

Lines 533-534: However, this elongated the sailing distance and resulted in more air pollutant emissions.

**Minor comments:**

**Q22. Line 86:**

Modify to "In addition, a scenario …"

**Response:**

Thank you. We have replaced "another scenario" with "a scenario".

**Revisions in Manuscript:**

Line 93: In addition,  **a** scenario without the DECA policy was performed to evaluate the effect of China's gradually implemented DECA policy, considering the actual change of interannual ship activities.

**Q23. Line 122:**

What is a "multidimensional analysis"?

**Response:**

Thank you. We have revised "multidimensional" to "from multiple perspectives".

**Revisions in Manuscript:**

Line 135-136: Finally, the ship emission inventory datasets will be established and used for visualization and  analysis **from multiple perspective**.

**Q24. Line 162:**

Replace "improvement" with "increase".

**Response:**

Thank you. This sentence has been removed (see **Q7** for revisions).

**Q25. Line 196.**

Skip "diagrammatic sketch".

**Response:**

Thank you. We have removed the words.

**Revisions in Manuscript:**

Line 229: Figure 4 illustrates  the ship route restoration algorithm, taking a segment of AIS positions as an example.

**Q26. Line 225 and several other places:**

Replace "implement" with "implementation".

**Response:**

Thank you. We have adopted the replacement throughout the manuscript.

**Q27. Line 314:**

What do you mean with "ship emission intensity"? Isn't this just daily emissions?

**Response:**

Thank you. We have removed the "intensity".

**Revisions in Manuscript:**

Line 368: The maximum daily ship emission  of $SO_2$ reached $6.4 \times 10^3$ Mg/day on September 22nd, 2018, 2.9 times of the lowest point, $2.2 \times 10^3$ Mg/day on January 1st, 2019.

**Q28. Line 335:**

What is meant with the "updating iteration speed of fleet"?

**Response:**

Thank you. The "updating iteration speed of fleet" was meant to express the "The

speed of vessel fleet renewal". We have deleted this expression as it might be redundant.

**Revisions in Manuscript:**

Lines 388-390: However, even though the newly-built, large-scale ships as well as ships using clean fuel oil are all taking an increasingly large part in emission structure,  the rising trend of NOx emission is not yet reversed.

**Q29. Line 350:**

Avoid expressions like "dramatically".

**Response:**

Thank you. It has been changed to "sharply"

**Revisions in Manuscript:**

Line 404: Fortunately, in 2019, when most rigorous DECA 2.0 policy was implemented, it is clearly illustrated in Fig. 9 that all ports' $SO_2$ emissions were  **sharply** reduced.

**Q30. Line 394 and line 472:**

I think "aggrandized" is not the right expression here.

**Response:**

Thank you. The "aggrandized" in Line 394 has been changed to "increased", and the sentence in Line 472 has been re-written.

**Revisions in Manuscript:**

1) Line 453: However, several ships detoured outside the scope of DECA 2.0 to save the cost on more expensive clean fuel oil, which further elongated the sailing distance and thus  **increased** emissions in farther maritime areas.
2) Lines 533-534: However, this elongated the sailing distance and resulted in more air pollutant emissions.

**Q31. Line 487:**

What do you mean with "complexity of registration and operation"?

**Response:**

Thank you. The "complexity" here referred to the fact that there were normally multiple countries involved for a particular ship, including the state of registration, ship owner and actual operator, etc. Thus, the shipping emission reduction might need multi-party cooperation.

**Revisions in Manuscript:**

Lines 547-549: Meanwhile, the international cooperation is also urgently called for to jointly control ship emissions due to ships' strong spatial mobility and the **intricate relations between the state of registration, ship owner and actual operator.**

**Q32. Author contributions:**

Initials are sometimes used in different order as in the list of authors. Who did the SEIM model runs? Who extended the model? What are "multiple analytical perspectives"?

**Response:**

Thank you for pointing out this. The section of "Author contributions" has been modified.

**Revisions in Manuscript:**

Lines 566-571: XW and WY contributed equally. XW extended the SEIM model and did the model runs. WY did the data analysis. XW and WY are responsible for writing the manuscript and figures and tables presented in this paper. ZL and FD provided valuable ideas on data analysis of this research. SZ and HX helped collect and clean the ship data. JZ assisted in the model development work. HL and KH provided guidance on the research and revised the paper. All authors contribute to the discussion and revision.

**Q33. References:**

Skip the test articles by Aman

**Response:**

Thank you. These articles have been removed.

**Q34. Figure 4b**

Figure 4b includes several spelling errors.

**Response:**

Thank you. We have corrected all spelling errors in Figure 4b.

**Revisions in Manuscript:**

[Figure]

**Figure 4: Diagrammatic sketch of the ship route restoration algorithm. (a) Sketch map of the route restoration algorithm with an example of route AB. (b) Algorithm flow chart of the example of route AB.**

**Q35. Figure 6**

Figure 6: the y-axis does not fit to the RV sulfur content, in particular after 2017/07.

**Response:**

Thank you. We have modified the y-axis and added the legends to Fig. 6.

**Revisions in Manuscript:**

[Figure]

**Figure 6: Evolution of sulfur content requirements for fuels in DECAs and inland rivers in China. The percentages refer to the sulfur content of the fuel. The italics refer to the operating mode constrained by DECA policy. The y-axis is unevenly distributed to show the standard of fuel sulfur content.**

**Q36. Figure 7**

Figure 7, b and c: blue bars for 2013 should be included in the legend. Does the 2013 bar represent the sum of OGV, CV and RV or are RV missing?

**Response:**

Thank you. We have added the legend for blue bars for 2013. It represent the sum of shipping emissions from OGV and CV, i.e., RV was excluded (Fu et al., 2017).

**Revisions in Manuscript:**

[Figure]

**Figure 7: Annual changes of (a) seaborne trade and ship emissions of (b) SO$_2$ and (c) NO$_x$ from 2016 to 2019. Data in (a) are collected from Chinese Statistical Yearbook (NBS, 2020). Emissions of 2013 are derived from our previous work for comparison (Fu et al., 2017).**

**References:**

*Fu, M., Liu, H., Jin, X., and He, K.: National- to port-level inventories of shipping emissions in China, Environmental Research Letters, 12, 114024, 10.1088/1748-9326/aa897a, 2017.*

**Q37. Figure 8**

Figure 8 might be skipped because of its low contribution to the scientific value of the paper.

**Response:**

Thank you for your suggestion. We have dropped the previous Fig. 8 and corresponding analysis in the manuscript (see **Q13**).

**Q38. Figure 13**

Replace "comparation" with "comparison"

**Response:**

Thank you. We have corrected this spelling error in Fig. 12 (previously Fig. 13).

**Revisions in Manuscript:**

Figure 12: Interannual spatial change of $NO_x$ and $SO_2$ emissions from ships over China from 2016 to 2019. Annual average spatial distribution **comparison** of $NO_x$ emission for (a) OGVs, (b) CVs and (c) RVs. Interannual variations of $NO_x$ and $SO_2$ emission in different geographic regions for (d) OGVs, (e) CVs and (f) RVs.

**Q39. Figure 14**

Mention that this refers to the 200 nm zone.

**Response:**

Thank you. We have added the description of "in inland rivers and the 200 Nm zone of China" in the legend of Fig. 13 (previously Fig. 14).

**Revisions in Manuscript:**

Figure 13: Monthly variation of ship $SO_2$ emissions i**n inland rivers and 200 Nm zone of China** under Base condition and No-DECA scenarios in 2016-2019. The Base condition refer to the real condition. The No-DECA scenario reflect the emission based on the real ship activities without the DECA policy.

**Q40. Figure 15**

The horizontal bars giving the changes in % should be aligned along a vertical zero-line.

**Response:**

Thank you. Figure 15 is designed to be presented in the form of a **waterfall chart** (https://en.wikipedia.org/wiki/Waterfall_chart). The waterfall chart was developed by McKinsey & Company. It is used to show the cumulative process of results by allowing the columns suspending in mid-air to represent the contribution of each positive and negative value to the total amount. In a waterfall chart, each of the suspended bar starts with the end value of its previous bar. In recent years, this chart has been widely applied to present the driving forces of emission reduction or air quality and health benefits (*Zhang et al., 2019; Wang et al., 2020; Milovanoff et al., 2020; Shan et al., 2021*).

In this case, the horizontal bar refers to the absolute emission change compared to the previous year, while the percentages refer to the change rate in emissions relative to the previous year. Therefore, we hope to keep the form of the waterfall chart, but added the axes and labels in this revised version.

**Revisions in Manuscript:**

[Figure]

Figure 14: Regional contributions to annual reduction SO$_2$ emissions from ships within 12 Nm of the baseline of China's territorial sea. The figures inside the blue bars refer to the annual emissions, and the percentages refer to the relative change of emissions due to total ship activity change in C-12 Nm region or the DECA policies in each region.

**References:**

Zhang, Q., Zheng, Y., Tong, D., Shao, M., Wang, S., Zhang, Y., Xu, X., Wang, J., He, H., Liu, W., Ding, Y., Lei, Y., Li, J., Wang, Z., Zhang, X., Wang, Y., Cheng, J., Liu, Y., Shi, Q., Yan, L., Geng, G., Hong, C., Li, M., Liu, F., Zheng, B., Cao, J., Ding, A., Gao, J., Fu, Q., Huo, J., Liu, B., Liu, Z., Yang, F., He, K., and Hao, J.: Drivers of improved PM2.5 air quality in China from 2013 to 2017, Proc Natl Acad Sci U S A, 116, 24463-24469, 10.1073/pnas.1907956116, 2019.

Wang, H., He, X., Liang, X., Choma, E. F., Liu, Y., Shan, L., Zheng, H., Zhang, S., Nielsen, C. P., Wang, S., Wu, Y., and Evans, J. S.: Health benefits of on-road transportation pollution control programs in China, Proc Natl Acad Sci U S A, 117, 25370-25377, 10.1073/pnas.1921271117, 2020.

Milovanoff, A., Posen, I. D., and MacLean, H. L.: Electrification of light-duty vehicle fleet alone will not meet mitigation targets, Nature Climate Change, 10, 1102-1107, 10.1038/s41558-020-00921-7, 2020.

Shan, Y., Ou, J., Wang, D., Zeng, Z., Zhang, S., Guan, D., and Hubacek, K.: Impacts of COVID-19 and fiscal stimuli on global emissions and the Paris Agreement, Nature Climate Change, 11, 200-206, 10.1038/s41558-020-00977-5, 2020.

---

## Author Response (AR2)

**Authors' Response:**

**Manuscript Title**: Annual changes in ship emissions around China under gradually promoted control policies from 2016 to 2019

**Discussion Link:** https://acp.copernicus.org/preprints/acp-2021-212/#discussion

**Response to Editor's Comments**

Comments to the Author:

The authors have done a good job in addressing the technical aspects of the reviewers comments and I would not regard any of these as needing to go out for a further round of review. However, the quality of English within this document is still in much need of improvement, to the extent that I worry that technical nuance may be lost or distorted if the copy editing process is relied on to rectify this. I therefore ask the authors to make an attempt to improve the quality of English before I make a final decision on this paper.

**Response:**

Thank you very much for your positive comments. Following your suggestion, we made our manuscript polished by 3 native English-speaking editors from a professional organization, Academic Journal Editing (AJE). The following is the Editing Certificate. All revisions are visible in the marked-up manuscript.

[Figure]

Besides adjustments requested by the Editor or Referees, please check your manuscript carefully for typos, missing co-authors and their affiliations, terminology, updates of data in tables, or updates of variables in equations. All these have to be clarified with the Editor and therefore have to be included before you submit your revised manuscript. Should your manuscript be finally accepted it will not be possible to include such rather substantial changes anymore when your manuscript is in final production (proofreading).

**Response:**

Thank you for your kind reminder. We have checked our manuscript carefully for all typos, co-authors and their affiliations, terminology, data in tables, and variables in equations.